# SYSTEM AWARE UNLEARNING ALGORITHMS: USE LESSER, FORGET FASTER

## ABSTRACT

Machine unlearning aims to provide privacy guarantees to users when they request deletion, such that an attacker who can compromise the system post-unlearning cannot recover private information about the deleted individuals. Previously proposed definitions of unlearning require the unlearning algorithm to exactly or approximately recover the hypothesis obtained by retraining-from-scratch on the remaining samples. While this definition has been the gold standard in machine unlearning, unfortunately, because it is designed for the worst-case attacker (that can recover the updated hypothesis and the remaining dataset), developing rigorous, and memory or compute-efficient unlearning algorithms that satisfy this definition has been challenging. In this work, we propose a new definition of unlearning, called *system aware unlearning*, that takes into account the information that an attacker could recover by compromising the system (post-unlearning). We prove that system-aware unlearning generalizes commonly referred to definitions of unlearning by restricting what the attacker knows, and furthermore, may be easier to satisfy in scenarios where the system-information available to the attacker is limited, e.g. because the learning algorithm did not use the entire training dataset to begin with. Towards that end, we develop an exact system-aware-unlearning algorithm that is both memory and computation-time efficient for function classes that can be learned via sample compression. We then present an improvement over this for the special case of learning linear classifiers by using selective sampling for data compression, thus giving the first memory and time-efficient *exact unlearning* algorithm for linear classification. We analyze the tradeoffs between deletion capacity, accuracy, memory, and computation time for these algorithms.

## 1 INTRODUCTION

In the era of large-scale machine learning (ML) models, which are often trained on extensive datasets containing sensitive or personal information, concerns surrounding privacy and data protection have become increasingly prominent (Yao et al., 2024). These models, due to their high capacity to memorize patterns in the training data, may inadvertently retain and expose information about individual data points (Carlini et al., 2021). This presents significant challenges in the context of privacy regulations such as the European Union's General Data Protection Regulation (2016) (GDPR), California Consumer Privacy Act (2018) (CCPA), and Canada's proposed Consumer Privacy Protection Act, all of which emphasize the "right to be forgotten." As a result, there is a growing need for methods that enable the selective removal of specific training data from models that have already been trained, a process commonly referred to as *machine unlearning* (Cao & Yang, 2015).

Machine unlearning addresses the need to remove data from a model's knowledge base without the need to retrain the model from scratch each time there is a deletion request, since this can be computationally expensive and often impractical for large-scale systems. The overarching objective here is to ensure that, post-unlearning, a model "acts" as if the removed data were never part of the training process (Sekhari et al., 2021a; Ghazi et al., 2023; Guo et al., 2019). Traditionally, this has been defined through notions of exact (or approximate) unlearning, wherein the model's hypothesis after unlearning should be identical (or probabilistically equivalent) to the model obtained by retraining from scratch on the entire data after removing just the deleted points. While such definitions offer rigorous guarantees even in the most pessimistic scenarios, they often impose stringent

requirements, limiting the practical applicability of machine unlearning. This is evidenced by a dire lack of exact/approximate unlearning algorithms beyond the simple cases of convex loss functions.

At the core of the unlearning problem lies a fundamental question: *What does it truly mean to "remove" a data point from a trained model? And more importantly, when we provide privacy guaranteed for deleted points against an outside observer/attacker, what information can this attacker reasonably possess?* The current definitions of exact/approximate unlearning take a worst-case perspective here and focus on the output hypothesis being indistinguishable from a retrained model (Sekhari et al., 2021a; Ghazi et al., 2023; Guo et al., 2019; Cherapanamjeri et al., 2024). However, this approach overlooks a key aspect of the unlearning problem—the observer and its knowledge of the system. In the real world, the feasibility and complexity of unlearning should depend on what the observer can access—be it the model parameters, data retained by the ML system in its memory, data ever encountered by the ML system etc. For instance, consider a learning algorithm that relies on only a fraction of its training dataset to generate its hypothesis and hence the ML system only stores this data. In such cases, unlearning a data point should intuitively be more straightforward. Even if the entire data in the memory of the system is compromised at some point, only the privacy of the stored points are at jeopardy as long as the learnt model does not reveal much about points that were not used by the model. Even if an observer/attacker has access to larger public data sets that might include parts of the data the system was trained on, in such a system we could expect privacy for data that the system does not use directly for building the model to be preserved. Conversely, if the algorithm utilizes the entire dataset and retains all information in memory, unlearning becomes far more challenging, potentially requiring retraining from scratch. This suggests that, in practice, the difficulty of unlearning is not solely determined by the learning algorithm but also by the observer's ability to detect traces of the removed data stored in the system or otherwise observed.

**Contributions.** We propose a new, *system-aware formulation of machine unlearning*, which incorporates the observer's perspective into the unlearning process. By explicitly considering what the observer knows about the system, we argue that exact unlearning, as traditionally defined, is often unnecessarily strict and computationally inefficient. Our framework leverages the fact that many ML systems do not depend on the entirety of their training data equally, allowing for more efficient and targeted unlearning approaches that better balance computational cost and privacy guarantees.

We then present a general-purpose, exact system-aware unlearning algorithm using data sharding for function classes that can learned using sample compression, establishing theoretical bounds on its computation time, memory requirements, deletion capacity, and excess risk guarantees. Previous works using data sharding for unlearning, such as Bourtoule et al. (2021), lack such theoretical guarantees. We also provide an improved system-aware unlearning algorithm for the special case of linear classification thus providing the first efficient *exact unlearning* algorithm for linear classification requiring sublinear in the number of samples. This is particularly noteworthy because under the traditional definition of unlearning, Cherapanamjeri et al. (2024) proved that exact unlearning for linear classification requires $\Omega(n)$ memory, essentially requiring the storage of the entire dataset.

Through this new lens on machine unlearning, we aim to bridge the gap between having rigorous theoretical guarantees and providing practical unlearning algorithms, thus hoping to develop scalable solutions for privacy-preserving machine learning (Tran et al., 2024; Cummings et al., 2023).

## 2 SETUP AND DEFINITION

Let $\mathcal{X}$ be the space of inputs, let $\mathcal{Y}$ be the space of outputs, let $\mathcal{P}$ be a distribution over an instance space $\mathcal{Z} = \mathcal{X} \times \mathcal{Y}$, let $\mathcal{F} \subseteq \mathcal{X}^{\mathcal{Y}}$ be a model class, and let $\ell : \mathcal{Y} \times \mathcal{Y} \to \mathbb{R}$ be a loss function. The goal of a learning algorithm is to take in a dataset $S \in \mathcal{Z}^*$ over the instance space and output a predictor $\widehat{f} \in \mathcal{F}$ which minimizes the excess risk compared to the best predictor $f^* \in \mathcal{F}$,

$$\mathsf{ExcessRisk}(\widehat{f}) \coloneqq \mathbb{E}_{(x,y)\sim\mathcal{P}}[\ell(\widehat{f}(x), y)] - \min_{f^*\in\mathcal{F}} \mathbb{E}_{(x,y)\sim\mathcal{P}}[\ell(f^*(x), y)].$$

Our goal in machine unlearning is to provide a privacy guarantee to data samples that request to be deleted, while ensuring that the updated hypothesis post-unlearning still has small excess risk. We first present the standard definition of machine unlearning, as stated in Sekhari et al. (2021b); Guo et al. (2019), often referred to as *certified machine learning*, which generalizes the commonly used *data deletion guarantee* from Ginart et al. (2019).

**Definition 1** (($\varepsilon, \delta$)-unlearning)**.** *For a dataset $S \in \mathcal{Z}^\star$, and deletions requests $U \subseteq S$, a learning algorithm $A : \mathcal{Z}^* \mapsto \Delta(\mathcal{F})$ and an unlearning algorithm $\bar{A} : \mathcal{Z}^* \times \mathcal{F} \times \mathcal{T} \mapsto \Delta(\mathcal{F})$ is ($\varepsilon, \delta$)-unlearning if for any $F \subseteq \mathcal{F}$,*

$$\Pr\left(\bar{A}(U, A(S), T(S)) \in F\right) \leq e^\varepsilon \cdot \Pr\left(\bar{A}(\varnothing, A(S \smallsetminus U), T(S \smallsetminus U)) \in F\right) + \delta,$$

*and*

$$\Pr\left(\bar{A}(\varnothing, A(S \smallsetminus U), T(S \smallsetminus U)) \in F\right) \leq e^\varepsilon \cdot \Pr\left(\bar{A}(U, A(S), T(S)) \in F\right) + \delta,$$

*where $T(S)$ denotes any intermediate auxiliary information that is available to $\bar{A}$ for unlearning.*

Sekhari et al. (2021a) also defined a notion of *deletion capacity*, which controls the number of samples that can be deleted while satisfying the above definition, and simultaneously ensuring good excess risk performance.

While the above definition, or its variations, have been the go-to definitions in machine unlearning research, we argue with a very simple example that it may, unfortunately, be an overkill even in some toy scenarios where we want to unlearn. Consider an algorithm that learns by first randomly sampling a small subset $C \subseteq S$ of size $m$ and then uses $C$ to train a model. Now, consider an unlearning algorithm that, when given some deletion requests $U$, simply retrains from scratch on $C \smallsetminus U$. Note that this is a valid unlearning algorithm from the perspective of an attacker who can only observe the model after unlearning because the model after unlearning contains no information about the deleted individuals $U$. On the other hand, this unlearning algorithm is not equivalent to rerunning the algorithm from scratch on $S \smallsetminus U$ which would involve sampling a different subset $C'$ of $m$ samples from $S \smallsetminus U$ and then training a model on $C'$. Since $C'$ contains $m$ samples whereas $C \smallsetminus U$ contains $m - |U|$ samples, the hypotheses learned using the respective datasets will likely not be statistically indistinguishable from each other. Thus, under Definition 1, this is not a valid unlearning algorithm, even though the above-mentioned attacker can gain no information about the deleted individuals.

The crucial thing to note is that Definition 1 considers a worst-case scenario that every point encountered by the unlearning algorithm except for the deletion requests, regardless of whether it is used or stored, are known to the attacker. However, a model trained on $C \smallsetminus U$ reveals no information about $U$ to an outside observer of the model after unlearning. In particular, samples that were never used for learning or stored in memory can never be leaked to the attacker. Unfortunately, previous definitions are unable to benefit from this aspect which is apparent from the lack of any non-trivial memory / compute efficient unlearning algorithms (Ghazi et al., 2023). However, before we provide a new definition of unlearning, we need to formalize the information that a learner can access about the system post-unlearning.

**Definition 2** (State-of-System)**.** *For an unlearning algorithm $A$, define the function $I_A : \mathcal{Z}^* \times \mathcal{Z}^* \mapsto \mathcal{Z}^*$ to denote the state of the system that is visible to an external observer post-unlearning. In particular, for any $S \subset \mathcal{Z}^*$, and deletion requests $U \subseteq S$, the quantity $I_A(S, U) \subseteq S$ is the subset of data points from dataset $S$ that is stored by or used in the output of the unlearning algorithm $A$ after $A$ has finished processing deletions requests $U$ after initially learning on $S$. This represents the information that an external observer/attacker gains about the original sample by observing the system after unlearning (e.g. the model, any stored samples, auxiliary data statistics, etc.).*

Whenever clear from the context, we will drop the subscript $A$ from $I_A$ to simplify the notation. For some examples of the state-of-system, for an unlearning algorithm that stores multiple models trained on different subsets of data, the state of the system denotes the union of the training data splits, and for an unlearning algorithm that upon a deletion request deletes every sample and returns a null predictor, the state of the system is the empty set. The adversary can access more information in the former scenario than the latter; thus, it should be more challenging to unlearn in the former.

**Definition 3** (System-Aware-($\varepsilon, \delta$)-Unlearning)**.** *Let $A : \mathcal{Z}^* \times \mathcal{Z}^* \mapsto \Delta(\mathcal{F})$ be a (possibly randomized) learning-unlearning algorithm, such that for a dataset $S$ and deletion requests $U$, $A(S, U)$ returns a hypothesis in $\mathcal{F}$ after first learning on sample $S$ and then processing a set of deletion requests $U$. We say that the algorithm $A$ is system-aware-($\varepsilon, \delta$)-unlearning if for all $S$ and $U \subseteq S$, there exists a $S'$ such that $I_A(S', \varnothing) = I_A(S, U)$ and $S' \cap U = \varnothing$, such that for all $F \subseteq \mathcal{F}$*

$$\Pr(A(S, U) \in F) \leq e^\varepsilon \cdot \Pr(A(S', \varnothing) \in F) + \delta$$

*and*

$$\Pr(A(S', \varnothing) \in F) \le e^{\varepsilon} \cdot \Pr(A(S, U) \in F) + \delta,$$

*where $I_A$ captures the state-of-the system after running $A$.*

System aware unlearning requires that the model output after initially learning on $S$ and then unlearning $U$ be indistinguishable from a model that learns on some plausible $S'$ from the perspective of the attacker and processes no deletion requests. Notice that $S'$ contains no information about $U$. Thus, we have properly unlearned if we can match the model and system state of the algorithm on $S'$. By taking $S' = S \smallsetminus U$, we recover the traditional notion of unlearning from Definition 1. Informally speaking, Definition 3 requires us to output a hypothesis that is statistically indistinguishable from retraining-from-scratch on a dataset that has no information about $U$. If an unlearning algorithm satisfies Definition 3 with $\varepsilon, \delta = 0$, then we say that the algorithm is an *exact system aware unlearning algorithm*.

**Why is only considering the system-state sufficient to provide privacy guarantees?** Consider when the unlearning algorithm $A$ satisfies $A(S, U) = f(I_A(S, U))$ for some fixed (possibly randomized) function $f$. This implies that $A(S, U) = A(S', \varnothing)$ since $I_A(S', \varnothing) = I_A(S, U)$, which means that Definition 3 is satisfied with $\varepsilon, \delta = 0$. Satisfying Definition 3 with $\varepsilon, \delta = 0$ implies that the Kullback–Leibler (KL) divergence between $\Pr(A(S, U) \mid I_A(S, U), U)$ and $\Pr(A(S', \varnothing) \mid I_A(S', \varnothing))$ is 0. Through the relationship between KL-divergence and mutual information along with $A(S, U) = A(S', \varnothing)$ and $I_A(S', \varnothing) = I_A(S, U)$, satisfying Definition 3 with $\varepsilon, \delta = 0$ implies that the conditional mutual information of $I(U; A(U, S) \mid I_A(S', \varnothing)) = 0$. This means that *given the state of the system after unlearning $I_A(S, U) = I_A(S', \varnothing)$, there is no mutual information between the deleted individuals $U$ and the output of the unlearning algorithm $A(S, U)$.* Thus, we simply need to ensure that the state of the system does not contain any information about the deleted individuals.

In the next section, we exploit the fact that algorithms that use or store fewer samples when training are easier to unlearn.

# 3 A SIMPLE APPROACH TO UNLEARNING FOR CORE SET ALGORITHMS VIA SHARDING

Since the attacker can only gain access to information stored by the system and used in the unlearned model, then we want to learn predictors that are dependent on a small number of samples. We formally define these type of algorithms as *core set based learning algorithm*.

**Definition 4** (Core Set Based Learning Algorithms). *A learning algorithm $\text{ALG}_{\text{CS}} : \mathcal{Z}^* \mapsto \mathcal{F}$ is said to be a core set-based learning algorithm if there exists a mapping $\mathfrak{C} : \mathcal{Z}^* \mapsto \mathcal{Z}^*$ such that for any $S \subseteq \mathcal{Z}$,*

$$\text{ALG}_{\text{CS}}(S) = \text{ALG}_{\text{CS}}(\mathfrak{C}(S)). \tag{1}$$

*We define $\mathfrak{C}(S)$ to be the core set of $S$.*

The output of $\text{ALG}_{\text{CS}}(S)$ only relies only on samples in $\mathfrak{C}(S)$. We can think of the core set $\mathfrak{C}(S \smallsetminus U)$ as the state of the system $I_A(S, U)$ and use the properties of core set algorithms to design exact unlearning algorithms. Many sample compression-based learning algorithms for classification tasks, such as SVM or selective sampling, are core set based learning algorithms (Hanneke & Kontorovich, 2021; Floyd & Warmuth, 1995). Additionally, the unlearning algorithms based on core set based learning algorithms are extremely fast because the deletion of a point outside the core set can be removed for free, so we only perform computation at the time of unlearning for a small number of points. We present a simple and fast unlearning algorithm (Algorithm 1) using core set based learning algorithms and data sharding to leverage the fact that samples which are not used or stored by the model are unlearned for free. Algorithm 1 is a general framework for system aware unlearning that applies to a variety of settings, including to non-convex function classes.

Algorithm 1 learns $K$ independent hypotheses using some suitable core set based learning algorithm $\text{ALG}_{\text{CS}}$. Each of the $K$ hypotheses is based on an independent core set $\mathfrak{C}(S^{(1)}), \ldots, \mathfrak{C}(S^{(K)})$. To process a set of deletion requests $U$, Algorithm 1 replaces the core sets containing points from $U$ with a core set that does not depend on $U$ at all and returns a hypothesis based on that core set.

---

**Algorithm 1** General purpose unlearning algorithm using sharding

---

**Input:** • Dataset $S$ of size $T$.
  • Deletion request $U \subseteq S$.
  • Core set deletion capacity $K$.
  • Core set-based learning algorithm $\text{ALG}_{\text{CS}}$.

1: **function** LEARNBYSHARDING( Dataset $S$, Deletion Capacity $K$)
2:      Partition $S$ into $K$ shards $S^{(1)}, \ldots, S^{(K)}$ uniformly at random.
3:      **for** $k \in [K]$ **do**
4:          $f^{(k)}, \mathfrak{C}(S^{(k)}) \leftarrow \text{ALG}_{\text{CS}}(S^{(k)})$, $f^{(k)}$ is the hypothesis and $\mathfrak{C}(S^{(k)})$ is the core set
5:      Define $\mathcal{T} = (\mathcal{T}_{\mathfrak{C}} = \{\mathfrak{C}(S^{(1)}), \ldots, \mathfrak{C}(S^{(K)})\}, \mathcal{T}_f = \{f^{(1)}, \ldots, f^{(K)}\})$
6:      **return** $\widehat{f}^* \leftarrow f^{(1)}$, and store $\mathcal{T} = (\mathcal{T}_{\mathfrak{C}}, \mathcal{T}_f)$.

7: **function** NEXTPRESERVEDPREDICTOR($z, \widehat{f}^*, \mathcal{T}$)
8:      Find a $\mathfrak{C}(S^{(j)})$ such that $\mathfrak{C}(S^{(j)}) \cap U = \varnothing$
9:      **if** no such $\mathfrak{C}(S^{(j)})$ exists **then**
10:         Replace each $\mathfrak{C}(S^{(i)}) \leftarrow \varnothing$ and $f^{(i)} \leftarrow \vec{0}$ and **return** $\widehat{f}^* \leftarrow \vec{0}$
11:      **else**
12:         **for** each $\mathfrak{C}(S^{(i)})$ **do**
13:            **if** $U \cap \mathfrak{C}(S^{(i)}) \neq \varnothing$ **then**
14:              Replace $\mathfrak{C}(S^{(i)})$ and $f^{(i)}$ with $\mathfrak{C}(S^{(j)})$ and $f^{(j)}$
15:         Swap $\mathfrak{C}(S^{(j)})$ and $f^{(j)}$ with $\mathfrak{C}(S^{(1)})$ and $f^{(1)}$, and then **return** $\widehat{f}^* \leftarrow f^{(1)}$

16: $\widehat{f}^*, \mathcal{T} \leftarrow$ LEARNBYSHARDING$(S, K)$           # Learn $K$ independent predictors on $S$
17: **return** NEXTPRESERVEDPREDICTOR$(\widehat{f}^*, \mathcal{T}, U)$    # Return a predictor untouched by deletion

---

Thus, we have $\mathsf{I}(S, U) = \mathcal{T}_{\mathfrak{C}}$, which is the remaining core sets in memory after learning on $S$ and then unlearning $U$. We prove that Algorithm 1 satisfies exact system aware unlearning.

**Theorem 1.** *For a given input dataset $S$, parameter $K \geq 1$ and deletion requests $U \subseteq S$, let $\mathfrak{C}^{(1)}, \ldots, \mathfrak{C}^{(K)}$ denote the remaining core sets in $\mathcal{T}$ after unlearning using Algorithm 1. Then, Algorithm 1 is an exact system-aware-unlearning algorithm (Definition 3 with $\varepsilon = \delta = 0$) with $S' = \mathfrak{C}^{(1)} \cup \cdots \cup \mathfrak{C}^{(K)}$.*

From the perspective of the attacker, the output after unlearning looks exactly the same as training a model on each of the core sets in $\mathcal{T}$ after unlearning because the only information stored in the system after unlearning are the $K$ core sets and the predictors trained on them.

We remark here that despite how simple this idea is, this unlearning algorithm is not captured by traditional definitions of unlearning in Definition 1, that requires the output after unlearning a sample $z_i$ to match the output of Algorithm 1 on the remaining dataset $S \setminus \{z_i\}$. If $z_i \in \mathfrak{C}(S^{(k)})$ for some $k$, we would have to update $f^{(k)}, \mathfrak{C}(S^{(k)})$ to match the output of $f^{(k)'}, \mathfrak{C}(S^{(k)})' \leftarrow \text{ALG}_{\text{CS}}(S^{(k)} \setminus \{z_i\})$ in order to unlearn $z_i$. However, note that $f^{(k)'}, \mathfrak{C}(S^{(k)})'$ could be very different from $f^{(k)}, \mathfrak{C}(S^{(k)})$ and updating the predictor could be very expensive. Under system aware unlearning, we can simply avoid this recomputation. Note that no computation needs to be done for Algorithm 1 at the time of unlearning, as we simply return a predictor that has been untouched by deletion.

We define the deletion capacity of an unlearning algorithm to be the number of deletions the algorithm can tolerate while maintaining a guarantee on the excess risk We define a core set deletion to be a deletion of point in $\mathfrak{C}(S)$. For core set algorithms, we are concerned with *core set deletion capacity*, the number of core set deletions an algorithm can tolerate, since deletions outside the core set do not affect the model. The algorithm designer specifies the desired bound $K$ on the core set deletion capacity, and Algorithm 1 divides the dataset into $K$ shards accordingly.

**Theorem 2.** *If the core set based learning algorithm $\text{ALG}_{\text{CS}}$ satisfies the excess risk bound,*

$$\mathbb{E}_{(x,y) \sim \mathcal{P}}[\ell(\widehat{f}(x), y)] - \min_{f^* \in \mathcal{F}} \mathbb{E}_{(x,y) \sim \mathcal{P}}[\ell(f^*(x), y)] \leq R(T, \delta),$$

*with probability at least $1 - \delta$ after learning on a dataset of size $T$. Then, after up to $K$ core set deletions, the excess risk of Algorithm 1 satisfies*

$$\mathbb{E}_{(x,y)\sim\mathcal{P}}[\ell(\widehat{f}(x),y)] - \min_{f^*\in\mathcal{F}}\mathbb{E}_{(x,y)\sim\mathcal{P}}[\ell(f^*(x),y)] \le R(T/K,\delta),$$

*with probability at least $1 - K\delta$. Let $\mathfrak{C}(S^{(i)})$ denote the expected size of the core set of $\mathrm{ALG_{CS}}$ on shard $S^{(i)}$. The memory required by Algorithm 1 for unlearning is $\sum_{i=1}^{K}|\mathfrak{C}(S^{(i)})|$.*

The proof of the above theorem is straightforward. Until each of the $K$ core sets has been hit with a deletion, Algorithm 1 can maintain excess error guarantees. We directly trade off core set deletion capacity at the cost of excess error rates. Note that we can delete all of the points outside of $\mathfrak{C}(S^{(i)})$ core sets without any impact on the core set deletion capacity. Furthermore, observe that $K$ bounds the worst case core set deletion capacity because deletions of multiple points within the same core set only decrease the deletion capacity by 1. After $K$ core set deletions, in expectation, $\frac{K}{e}$ of the shards remain untouched, where $e$ is the universal mathematical constant. We emphasize that the unlearning guarantee continues to be met even after the core set deletion capacity is exhausted.

The memory required for unlearning scales with the core set deletion capacity $K$. Note that for many core set algorithms, such as selective sampling or SVM, the size of the core set can be exponentially smaller than the size of $S$ (Cortes & Vapnik, 1995; Dekel et al., 2012; Shalev-Shwartz & Ben-David, 2014; Feldman, 2020).

# 4 BETTER UNLEARNING ALGORITHMS VIA SELECTIVE SAMPLING: THE CASE STUDY OF LINEAR CLASSIFICATION

Using sharding is a good generic starting point for unlearning, but can we improve upon some of the tradeoffs of sharding using a different technique? In this section, we show that for linear classification, we can use selective sampling to design an exact unlearning algorithm that demonstrates better tradeoffs between deletion capacity, memory requirements, and excess error compared to sharding, thus resulting in the first space and time efficient exact unlearning algorithm for linear classification.

Selective sampling (Cesa-Bianchi et al., 2009; Dekel et al., 2012; Zhu & Nowak, 2022; Sekhari et al., 2023; Hanneke et al., 2014) is the problem of finding a classifier with low error while only using the label of very few points and has become particularly important as datasets become larger and labeling them becomes more expensive. Selective sampling algorithms only query the label of points whose label they are uncertain of and only update the model on points that they query. Furthermore, unqueried points are never stored in memory and never used in learning. Selective sampling is a core set based learning algorithm where the core set is exactly the set of queried points.

Linear classification is a fundamental learning problem in both theory and practice. While it is a useful theoretical primitive in algorithm design, this simple problem also has relevance for practice, for example, in large foundation models and generative models, the last layers of these models are often fine-tuned using linear probing, which trains a linear classifier on representations learned by a deep neural network (Belinkov, 2022; Kornblith et al., 2019). As unlearning gains increasing attention for these large-scale ML models, we hope that the following improvements for unlearning linear classification will find practical applications.

**Assumptions.** We consider the problem of binary linear-classification. Let $x \in \mathbb{R}^d$ be such that $\|x\| \le 1$ and $y \in \{+1, -1\}$. Furthermore, we assume that there exists a $\mathbf{u} \in \mathbb{R}^d$, $\|\mathbf{u}\| < 1$ such that $\mathbb{E}[y_t \mid x_t] = \mathbf{u}^\top x_t$. Also known as the realizability assumption for binary classification, this ensures that the Bayes optimal predictor for $y_t$ is $\mathrm{sign}(\mathbf{u}^\top x_t)$. Our goal in linear-classification is to find a hypothesis that performs well under $0 - 1$ loss, i.e. set $\ell(f(x),y) = \mathbb{1}\{f(x) \ne y\}$. With this goal in mind, we define the excess risk for a hypothesis $w$ as

$$\mathsf{ExcessRisk}(w) := \mathbb{E}_{(x,y)\sim\mathcal{P}}\big[\mathbf{1}\{\mathrm{sign}(w^\top x) \ne y\} - \mathbf{1}\{\mathrm{sign}(\mathbf{u}^\top x) \ne y\}\big]. \tag{2}$$

We use the selective sampling algorithm BBQSAMPLER from Cesa-Bianchi et al. (2009) to design the unlearning algorithm. Algorithm 2 uses the BBQSAMPLER to learn a predictor that only depends on a small number of core set points, where $\mathfrak{C}(S) = \mathcal{Q}$. Note that the last predictor returns an ERM over $\mathfrak{C}(S)$. Then when unlearning $U$, we update the predictor to be an ERM over $\mathfrak{C}(S) \setminus U$ and

---

**Algorithm 2** Unlearning algorithm for linear classification using selective sampling

---

**Input:** • Dataset $S$ of size $T$
- Deletion request $U$
- Deletion capacity $K > 0$
- Sampling parameter $0 \le \kappa \le 1$

1: **function** BBQSAMPLER($S, K, \kappa$)
2:      Set regularization $\lambda = K$
3:      Initialization: $w_0 = 0, A_0 = \lambda I, b_0 = \vec{0}, \mathcal{Q} = \varnothing$
4:      **for** each $t = 1, 2, \ldots, T$ **do**
5:          Observe instance $x_t$
6:          **if** $x_t^\top A_{t-1}^{-1} x_t > T^{-\kappa}$ **then**        # Only update the predictor on queried points
7:              Query label $y_t$, and update $\mathcal{Q} = \mathcal{Q} \cup \{(x_t, y_t)\}$.
8:              Update $A_t \leftarrow A_{t-1} + x_t x_t^\top, b_t \leftarrow b_{t-1} + y_t x_t, w_t \leftarrow A_t^{-1} b_t$.
9:          **else**
10:            Set $A_t \leftarrow A_{t-1}, b_t \leftarrow b_{t-1}, w_t \leftarrow w_{t-1}$
11:      **return** $\mathcal{Q}, A_T, b_T, w_T$

12: **function** DELETIONUPDATE($\mathcal{Q}, X, b, w, U$)
13:      **for** $(x, y) \in U$ such that $(x, y) \in \mathcal{Q}$ **do**
14:          Define $\mathcal{Q} = \mathcal{Q} \smallsetminus \{x\}$
15:          Update $X \leftarrow X - xx^\top, b \leftarrow b - yx$ and $w \leftarrow A^{-1} b$.
16:      **return** $\mathcal{Q}, X, b, w$

17:
18: $\mathcal{Q}, X, b, w \leftarrow$ BBQSAMPLER($S, \lambda, \kappa$)        # Learn a predictor via selective sampling
19: $\mathcal{Q}, X, b, w \leftarrow$ DELETIONUPDATE($\mathcal{Q}, X, b, w, U$)    # Update the predictor for core set deletions
20: **return** $\text{sign}(w^\top x)$

---

remove $U$ from memory. After unlearning, the model output and everything stored in memory only relies on $\mathfrak{C}(S) \smallsetminus U$.

**Theorem 3.** *Let $\mathfrak{C}(S)$ denote the core set of the* BBQSAMPLER *on sample $S$. Algorithm 2 is an exact system-aware-unlearning algorithm (3) with $S' = \mathfrak{C}(S) \smallsetminus U$.*

The proof relies on a key attribute of the BBQSAMPLER - its monotonic query condition with respect to deletion. If the BBQSAMPLER is executed on $S$ and then re-executed on $S$ with some point $x_j$ removed, every $x_t$ which was queried before $x_j$ was removed will still be queried after $x_j$ is removed.

**Lemma 1.** *The query condition from Algorithm 2 is monotonic with respect to deletion. Specifically, if $x_t^\top A_t^{-1} x_t > T^{-\kappa}$, then $x_t^\top A_{t \smallsetminus x_j}^{-1} x_t > T^{-\kappa}$ for any $j \in [T]$ such that $j \neq t$.*

The query condition of the BBQSAMPLER is only $x$-dependent and does not depend on the labels $y$ at all. In particular, we query the label on $x_t$ if the direction containing $x_t$ is not well sampled. The monotonicity of the query condition is evident from the fact that if a direction was not well sampled before deletion, it will also not be well-sampled if some previous samples were deleted.

This monotonicity is a unique feature of the BBQSAMPLER. Other selective sampling algorithms, such as ones from Dekel et al. (2012) or Sekhari et al. (2023), use a query condition that depends on the labels $y$ of previously seen points. Due to the noise in these $y$'s, $y$-dependent query conditions are not monotonic; points that were queried can become unqueried. This makes it difficult and expensive to compute the core set after unlearning. We note that since the BBQSAMPLER uses a $y$-independent query condition, it is suboptimal in terms of excess error before unlearning compared to algorithms from Dekel et al. (2012) or Sekhari et al. (2023). However, we are willing to tolerate a small increase in the excess error in order to unlearn efficiently. Additionally, it is unclear how much the error of $y$-dependent selective sampling algorithms would suffer after a core set deletion.

Using the monotonic query condition, we see that $\mathfrak{C}(\mathfrak{C}(S) \smallsetminus U) = \mathfrak{C}(S) \smallsetminus U$, so we do not need to re-execute the BBQSAMPLER at the time of unlearning in order to determine the new set of queried points. We can simply remove the effect of $U$ on the predictor, and we only need to make an update for deletion requests in $U$ that are also in $\mathfrak{C}(S)$.

**Why is Algorithm 2 not a valid unlearning algorithm under the prior unlearning definition (Definition 1)?** When a queried point is deleted, an unqueried point could become queried. Thus, we have $\mathfrak{C}(S \smallsetminus U) \neq \mathfrak{C}(S) \smallsetminus U$. Thus, under traditional notions of exact unlearning, during DELETIONUPDATE, not only would we have to remove the effect of $U$, but we would also have to add in any unqueried points that would have been queried if $U$ never existed in $S$. Additionally, it is computationally inefficient to determine which points would have been queried and unnecessary from a privacy perspective. An attacker could never have known that such an unqueried point existed and should have become queried after deletion since it was never used in the original model.

We next bound the memory requirement for *Algorithm 2* and show that the predictor after unlearning maintains low excess risk.

**Theorem 4.** *With probability at least $1 - \delta$, the excess risk of the predictor $\tilde{w}$ in Algorithm 2 is bounded by*

$$\mathsf{ExcessRisk}(\widetilde{w}) = O\left(\min_{\varepsilon}\left\{\frac{T_\varepsilon}{T} + \frac{dT^\kappa \log T}{T} + \frac{\log(1/\delta)}{T}\right\}\right),$$

*even after unlearning*

$$K = O\left(\frac{\bar{\varepsilon}^2 \cdot T^\kappa}{d \log T}\right)$$

*many core-set deletions, where $T_\varepsilon = \sum_{t=1}^{T} \mathbf{1}\{|\mathbf{u}^\top x_t| \leq \varepsilon\}$, and $\bar{\varepsilon}$ denotes the minimizing $\varepsilon$ in the excess risk bound above. Furthermore, the memory required by Algorithm 2 is determined by the number of core set points which is bounded by $|\mathfrak{C}(S)| \leq O(dT^\kappa \log T)$.*

We remark that $T_{\bar{\varepsilon}}$ represents the number of points where even the Bayes optimal predictor is unsure of the label, which we expect to be small in realistic scenarios. We give a proof sketch of the theorem. The bound on the query complexity of the BBQSAMPLER before unlearning is well known and can be derived using standard analysis for selective sampling algorithms from Cesa-Bianchi et al. (2009). The number of queries made by the BBQSAMPLER exactly bounds the number of points in the core set. To bound the excess risk, we first show that the final predictor $\hat{w} = w_T$ from the BBQSAMPLER correctly classifies all of the unqueried points outside of the $T_{\bar{\varepsilon}}$ margin points. Let $\tilde{w}$ be the predictor after $K$ core set deletions. We want to ensure that the sign of $\hat{w}$ and the sign $\tilde{w}$ remain the same for all the unqueried points. We do so by first demonstrating that $\hat{w}$ exhibits stability (Bousquet & Elisseeff, 2002; Shalev-Shwartz et al., 2010) on any unqueried point $x$, $|\hat{w}^\top x - \tilde{w}^\top x| < \sqrt{K \cdot d \log T \cdot T^{-\kappa}}$. Then we show that $\hat{w}$ has a $\bar{\varepsilon}/2$ margin on the classification of every unqueried point. Putting these together, we show that for up to $K \leq O\left(\frac{\bar{\varepsilon}^2 \cdot T^\kappa}{d \log T}\right)$ deletions, we can ensure that the sign of $\hat{w}$ and $\tilde{w}$ on the unqueried points is the same. Thus, we can maintain correct classification on unqueried points. We cannot make any guarantees on the $|\mathfrak{C}(S)|$ queried points and the $T_{\bar{\varepsilon}}$ margin points, so we assume full classification error on those points. Finally, we use techniques from Bousquet et al. (2004) to convert the empirical classification loss to an excess risk bound.

**Memory required for unlearning.** The memory required for unlearning is exactly the number of core set points, $O(dT^\kappa \log T)$. Unlike sharding, the memory does not scale with the core set deletion capacity. Under system aware unlearning, we obtain the first exact unlearning algorithm for linear classification which uses sublinear memory and does not need to store the entire dataset.

**Deletion capacity and error rates.** Theorem 4 bounds the core set deletion capacity. Since $\kappa$ is a free parameter, we can tune it to increase the core set deletion capacity at the cost of increasing the excess risk after deletion. We are trading off deletion capacity at the cost of performance.

**Lemma 2.** *If the underlying data generating distribution has a hard margin of $\gamma$, i.e. there exists a $\gamma$ such that $T_\gamma = 0$. Appropriately tuning $\kappa$ in Theorem 4, we get that, for any $p \in (0, 1)$, Algorithm 2 can tolerate up to $K = O(\gamma^2 \cdot T^{1-p})$ deletions while ensuring that the excess risk is $O(\frac{1}{T^p})$.*

### 4.1 COMPARISON TO SHARDING

The sharding technique from Algorithm 1 is a great out-of-the-box strategy for unlearning that can be applied to general function classes and the agnostic setting. However, for linear classification under the mean realizability assumption, Algorithm 2 demonstrates better tradeoffs between deletion capacity, memory, and excess risk.

We compare the tradeoff between excess risk and core set deletion capacity for Algorithm 2 described in Lemma 2 to the tradeoff between excess risk and core set deletion capacity for Algorithm 1 using sharding and selective sampling on each shard, where $\text{ALG}_{\text{CS}}$ to be the optimal selective sampling algorithm for linear classification from Dekel et al. (2012). As in Lemma 2, we assume a hard margin of $\gamma$. The excess risk bound of the optimal selective sampling algorithm on a dataset $S$ of size $T$ is

$$\text{ExcessRisk}(q) \leq O\left(\frac{d\log T + \log(T/\delta)}{\gamma T} + \frac{\log(\log T/\delta)}{T}\right),$$

derived with a standard online-to-batch conversion where $q$ is a randomly selected predictor from $\{w_1, \ldots, w_T\}$ (Dekel et al., 2012).

When the deletion capacity is set to $K = \gamma^2 \cdot T^p$, we plug in $T/K$ for $T$ in the bound above to get that the excess risk of Algorithm 1 after up to $K$ deletions is at most

$$\text{ExcessRisk}(q) = O\left(\frac{\gamma(d\log T + \log(T/\delta))}{T^p}\right).$$

Compare this to the excess bound of $\frac{1}{T^p}$ for Algorithm 2 for the same number of deletions. As $d$ and $T$ increase, Algorithm 2 can achieve a smaller regret bound for the same number of deletions of queried points in comparison to sharding.

Algorithm 2 also requires significantly less memory for unlearning compared to sharding. The memory required by Algorithm 2 is only $T^{1-p}$, while the memory required by sharding is $T^{1-p} \cdot d^2 \log^2 T$ (the deletion capacity $K = \gamma^2 \cdot T^p$ times the query complexity $N_T = \frac{d^2 \log^2 T}{\gamma^2}$ of the optimal selective sampling algorithm from Dekel et al. (2012)). Furthermore, since sharding uses a larger number of queried points, the probability of a queried point being deleted under sharding is greater than the probability under Algorithm 2; therefore, we would exhaust the deletion capacity quicker under sharding.

## 4.2 Expected Deletion Capacity

Notice that the deletion capacity of $K$ only applies to core set deletions. Assume that deletions are drawn without replacement from $\mu : \mathcal{X} \to [0, 1]$, a probability weight vector over the samples in $S$. This implies that probability of $x$ requesting for deletion, i.e. $\mu(x)$, only depends on $x$ and not on its index within $S$ or on other samples. This assumption is useful for capturing scenarios where the users make request for deletions solely based on their own data and have no knowledge of where in the sample they appear. We define $K_{\text{TOTAL}}$ as the total number of deletions we can process under $\mu$ before we exhaust the core set deletion capacity of $K$ and lose excess risk guarantees.

**Theorem 5.** *Consider any core-set algorithm $A$. Let $\pi$ denote denote a uniformly random permutation of the samples in $S$, and let $\sigma$ be a sequence of deletion requests samples from $\mu$, without replacement. Further, let $K_{\text{CSD}}$ denote the number of core set deletions within the first $K_{\text{TOTAL}}$ deletion requests, then for any $K \geq 1$,*

$$\Pr_{S,\pi,\sigma}(K_{\text{CSD}} > K) \leq \frac{1}{K}\mathbb{E}_S\left[\sum_{t=1}^{T}\mathbb{E}_\pi[\mathbf{1}\{x_t \in \mathfrak{C}_A(\pi(S))\}] \cdot \sum_{k=1}^{K_{\text{TOTAL}}}\mathbb{E}_\sigma[\mathbf{1}\{x_t = x_{\sigma_k}\}]\right].$$

*where $\mathfrak{C}_A(\pi(S))$ denotes the coreset resulting from running $A$ on the permuted dataset $\pi(S)$. Instantiating the above bound for Algorithm 2 implies that*

$$\Pr_{S,\pi,\sigma}(K_{\text{CSD}} > K) \leq \frac{K_{\text{TOTAL}} \cdot T^\kappa}{K}\mathbb{E}_S[\mathbb{E}_{x\sim\mu}[x^\top \overline{M} x]]$$

*where $\overline{M} := \mathbb{E}_\pi\left[\frac{1}{T}\sum_{s=1}^{T} A_{s-1}^{-1}\right]$ and $\kappa \in (0, 1)$ denotes the parameter for Algorithm 2.*

For a given deletion distribution $\mu$, Theorem 5 can be used to derive a bound on the number of deletions $K_{\text{TOTAL}}$ while ensuring that the probability of exhausting the core set deletion capacity is small. The bound on $K_{\text{TOTAL}}$ depends inversely on $\mathbb{E}_S[\mathbb{E}_{x\sim\mu}[x^\top \overline{M} x]]$; when $\mathbb{E}_S[\mathbb{E}_{x\sim\mu}[x^\top \overline{M} x]]$ is small, Algorithm 2 can tolerate a large number of deletions $K_{\text{TOTAL}}$ before exhausting its core

set deletion capacity $K$. $\mathbb{E}_S[\mathbb{E}_{x\sim\mu}[x^\top \overline{M} x]]$ can be interpreted as the expected value of the query condition $x^\top A_t^{-1} x$ when Algorithm 2 encounters $x$ during its execution where $x$ is drawn from the deletion distribution. The query condition decreases as it encounters and queries more points. Thus, $\mathbb{E}_S[\mathbb{E}_{x\sim\mu}[x^\top \overline{M} x]]$ is decreasing as $T$ increases, and we would expect it to be small for large $T$. $x^\top \overline{M} x$ is maximized when $x$ lies in a direction which does not occur very often. Deletion distributions $\mu$ which place a lot of weight on poorly sampled directions will maximize $\mathbb{E}_S[\mathbb{E}_{x\sim\mu}[x^\top \overline{M} x]]$ and lead to smaller $K_{\text{TOTAL}}$. Given the deletion distribution, we can derive exact bounds for $\mathbb{E}_S[\mathbb{E}_{x\sim\mu}[x^\top \overline{M} x]]$ which lead to bounds on $K_{\text{TOTAL}}$.

**Lemma 3.** *Let the deletion distribution $\mu$ be the uniform distribution. Working out the bound from Theorem 5, we have*

$$\Pr_{S,\pi,\sigma}(K_{\text{CSD}} > K) \le \frac{K_{\text{TOTAL}} \cdot T^\kappa \cdot d\log T}{K \cdot T}.$$

*We can process a total of*

$$K_{\text{TOTAL}} = \frac{c \cdot K \cdot T}{dT^\kappa \log T}$$

*deletions such that the probability that we exhaust the core set deletion capacity of $K$ is at most $c$.*

### 4.3 EXPECTED DELETION TIME

We can make a similar argument for the deletion time. At the time of unlearning, we only need to make an update for deletions of points in the core set. For all other points, there is no computation time for unlearning. For a given $K_{\text{TOTAL}}$, the total number of deletions we can process under $\mu$ before we have exhausted the core set deletion capacity of $K$, which can be derived using Theorem 5, we can give an expression for the expected time for deletion.

**Theorem 6.** *For a deletion distribution $\mu$, if a core set algorithm $A$ can tolerate up to $K_{\text{TOTAL}}$ deletions before exhausting the core set deletion capacity $K$,*

$$\mathbb{E}[\text{time per deletion}] \le \frac{K}{K_{\text{TOTAL}}} \times \{\text{update time for a core set deletion}\}.$$

For Algorithm 2 under a uniform deletion distribution, we have

$$\mathbb{E}[\text{time per deletion}] \le \frac{d^3 T^\kappa \log T}{T},$$

by plugging in $K_{\text{TOTAL}}$ from Lemma 3 and using the fact that updating the predictor after the deletion of a core set point takes $O(d^2)$ time using the Sherman-Morrison update (Hager, 1989).

**Remark 1.** *For large $d$, the update time can be replaced by a quantity that depends on the eigenspectrum of the data's Gram matrix. Furthermore, since Algorithm 2 updates an ERM on $\mathfrak{C}(S)$ to an ERM on $\mathfrak{C}(S) \setminus \{z\}$, we can use gradient descent which takes $O(d)$ time per update.*

**Experiments.** We perform some toy experiments on unlearning for linear classification with Algorithm 2 in Appendix A.1. Our experiments show that Algorithm 2 can maintain low excess risk far beyond the core set deletion capacity derived in Theorem 4 even under worst case deletion schemes.

## 5 CONCLUSION

We proposed a new definition for unlearning, called system aware unlearning, that takes into account the information about the sample $S$ that could be leaked to an attacker who compromises the system, and we developed exact system aware unlearning algorithms based on sample compression learning algorithms. In particular, we used selective sampling to design a memory and time efficient unlearning algorithm for linear classification. It would be interesting to explore whether or not this analysis can be extended to general function classes and prove that function classes with finite eluder dimension (Russo & Van Roy, 2013) lead to memory and computation efficient exact system aware unlearning algorithms. Beyond exact unlearning algorithms, it would be interesting to explore how allowing for approximate system aware unlearning ($\varepsilon, \delta \neq 0$) can lead to even faster and more memory-efficient unlearning algorithms. Furthermore, we believe that accounting for the information that an attacker could have access to is an interesting direction to explore in generic privacy, beyond unlearning.

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

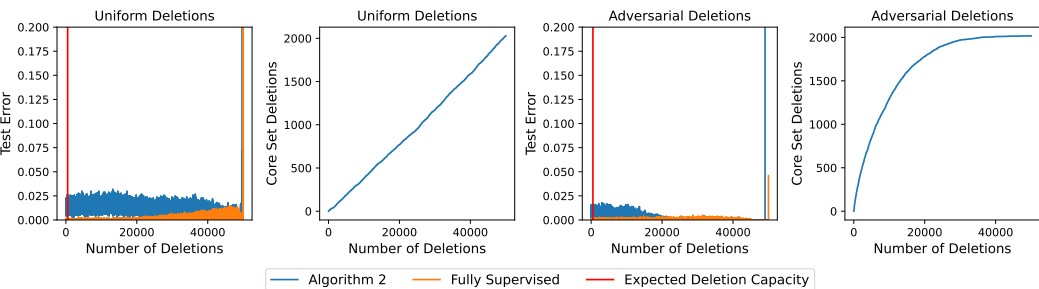

Figure 1: The first and third plots measure the test error of Algorithm 2 (selective sampling) over the course 50,000 deletions compared to the test error of retraining a fully supervised algorithm after each deletion under two different deletion schemes. The second and fourth plots graph the number of deleted core set points over the course of the 50,000 deletions under the two deletion schemes.

# A APPENDIX

## A.1 EXPERIMENTS

Our theoretical results provide guarantees for the worst case deletions. We experimentally verify our theory, and we demonstrate that the deletion capacity and error rates after unlearning for Algorithm 2 are much better in practice. We randomly generate 50,000 points in dimension $d = 100$ with a hard margin condition of $\gamma = 0.1$. We learn a classifier on these 50,000 points and process 50,000 deletions using Algorithm 2. After each deletion, we compare the test error of the classifier after unlearning from Algorithm 2 to test error of a fully supervised linear classification algorithm which learns on all of the undeleted points in the sample, including points which are unqueried and thus never used by Algorithm 2. The test error of the fully supervised algorithm represents the best possible error Algorithm 2 could hope to achieve after deletion. Note that Algorithm 2 can maintain comparable test error with the fully supervised predictor while only using ~ $4\%$ of the samples.

We test two different deletion schemes:

- *Uniform deletions*: Each deletion request is selected uniformly at random. This is to illustrate the case when the deletion distribution does not correlate at all with the query condition.
- *Adversarial deletions with respect to queried points*: Deletions are specifically selected in an attempt to maximize $x_t^\top A_t^{-1} x_t$. However, when a user requests for deletion, that user does not have knowledge of which individuals were queried or where their position in the sample is, so they cannot exactly calculate $x_t^\top A_t^{-1} x_t$, but the user may have some knowledge of the data distributions. We simulate this knowledge by deleting in decreasing order of $x_t^\top A_T^{-1} x_t$ where $A_T = I + \sum_{t=1}^{T} x_t x_t^\top$ represents the covariance matrix of the sample. This is to illustrate the case when the deletion distribution happens to correlate well with the query condition.

We observe that Algorithm 2 can maintain low classification error until essentially all of the points in the sample are deleted. Note that Algorithm 2 can maintain comparable test error compared to the fully supervised algorithm despite only using a fraction of the points. From Figure 1, we see that we can maintain low classification error far past the deletion capacity bound derived in Theorem 4 (around $1\%$ of points in this case), under both deletion schemes. This is particularly noteworthy for the adversarial deletion scheme which is designed to delete as many queried points as soon as possible which should quickly deteriorate the error Algorithm 2 since it only relies on queried points.

## A.2 OTHER RELATED WORK

Chourasia & Shah (2023) proposes a data deletion definition under adaptive requesters which does not require indistinguishability from retraining from scratch. They require that the model after deletion be indistinguishable from a randomized mapping $\pi$ on $S$ with $z_i$ replaced. This assumes that the attacker does not have knowledge of the unlearning algorithm itself. If the data deletion requesters are non-adaptive, then $\pi$ can be replaced by the unlearning algorithm $A$, but in general, system aware unlearning does not generalize this definition. The data deletion definition under

adaptive requesters makes the stronger assumption that the unlearning algorithm uses the entire sample to unlearn, but the weaker assumption that the attacker does not know the learning algorithm.

Beyond unlearning definitions, there has been much work in the development of unlearning algorithms. The current literature generally falls into two categories: exact unlearning algorithms which exactly reproduce the model from retraining from scratch on $S \setminus \{z_i\}$ (Ghazi et al., 2023; Cherapanamjeri et al., 2024; Bourtoule et al., 2021; Cao & Yang, 2015; Chowdhury et al., 2024) or approximate unlearning algorithms which use ideas from differential privacy (Dwork et al., 2014) to probabilistically recover a model that is "essentially indistinguishable" from the model produced from retraining from scratch on $S \setminus \{z_i\}$ (Izzo et al., 2021; Sekhari et al., 2021a; Chien et al., 2024). The exact unlearning algorithms are typically memory intensive and require the storage of the entire dataset and multiple models, while the approximate unlearning algorithms are much more memory efficient. Furthermore, existing lower bounds prove that there exist model classes with finite VC and Littlestone dimension where traditional exact unlearning requires the storage of the entire dataset (Cherapanamjeri et al., 2024). For large datasets, this makes exact unlearning under the traditional definition impractical. We prove that we can design practical exact system aware unlearning algorithms for linear classification which require sublinear memory in the number of samples.

## A.3 NOTATION

- $[n] = \{1, 2, \ldots, n\}$
- $A_T = \lambda I + \sum_{t=1}^T x_t x_t^\top$
- $A_{T \setminus U} = \lambda I + \sum_{t=1}^T x_t x_t^\top - \sum_{x_i \in U} x_i x_i^\top$ where $U$ is a set of deletions
- $A_{t \setminus x_j} = \begin{cases} \lambda I + \sum_{t=1}^T x_t x_t^\top - x_j x_j^\top & \text{when } j \leq t \\ \lambda I + \sum_{t=1}^T x_t x_t^\top & \text{otherwise} \end{cases}$
  for some $j \in [T]$
- $A_S = \lambda I + \sum_{x_t \in S} x_t x_t^\top$ where $S$ is a set of points
- $b_T = \sum_{t=1}^T y_t x_t$
- $b_T = \sum_{t=1}^T y_t x_t - \sum_{x_i \in U} y_i x_i$ where $U$ is a set of deletions
- $w_T = A_T^{-1} b_T$
- $w_{T \setminus U} = A_{T \setminus U}^{-1} b_{T \setminus U}$ where $U$ is a set of deletions
- $\|u\|_X = u^\top X u$, where $u \in \mathbb{R}^d$ and $X \in \mathbb{R}^{d \times d}$

## A.4 PROOFS FROM SECTION 3

**Theorem 1.** *For a given input dataset $S$, parameter $K \geq 1$ and deletion requests $U \subseteq S$, let $\mathfrak{C}^{(1)}, \ldots, \mathfrak{C}^{(K)}$ denote the remaining core sets in $\mathcal{T}$ after unlearning using Algorithm 1. Then, Algorithm 1 is an exact system-aware-unlearning algorithm (Definition 3 with $\varepsilon = \delta = 0$) with $S' = \mathfrak{C}^{(1)} \cup \cdots \cup \mathfrak{C}^{(K)}$.*

**Proof.** Let $\{\mathfrak{C}^{(1)}, \ldots, \mathfrak{C}^{(K)}\}$ be the core sets in $\mathcal{T}$ after unlearning, and let $\{f^{(1)}, \ldots, f^{(K)}\}$ be the predictors after unlearning. Define $S' = \mathfrak{C}^{(1)} \cup \cdots \cup \mathfrak{C}^{(K)}$. We have $S' \cap U = \varnothing$ by the way we update the core sets. The $K$ shards of $S'$ are exactly $\mathfrak{C}^{(\alpha_1)}, \ldots, \mathfrak{C}^{(\alpha_K)}$. $A(S', \varnothing)$ will execute $\text{ALG}_{\text{CS}}(\mathfrak{C}^{(\alpha_i)})$ on all $i$ shards. Since $\text{ALG}_{\text{CS}}$ is a core set based learning algorithm, this means executing $\text{ALG}_{\text{CS}}$ on each shard exactly leads to the predictors $\{f^{(1)}, \ldots, f^{(L)}\}$ and core sets $\{\mathfrak{C}^{(1)}, \ldots, \mathfrak{C}^{(K)}\}$ stored in memory. Thus, $S'$ satisfies $\mathsf{I}(S', \varnothing) = \mathsf{I}(S, U) = \mathcal{T}$. Both $A(S, U)$ and $A(S', \varnothing)$ return $f^{(1)}$ as the predictor; thus, we have $A(S, U) = A(S', \varnothing)$ which means Algorithm 2 satisfies exact system aware unlearning. $\square$

## A.5 PROOFS FROM SECTION 4

**Lemma 1.** *The query condition from Algorithm 2 is monotonic with respect to deletion. Specifically, if $x_t^\top A_t^{-1} x_t > T^{-\kappa}$, then $x_t^\top A_{t \setminus x_j}^{-1} x_t > T^{-\kappa}$ for any $j \in [T]$ such that $j \neq t$.*

**Proof.** Consider a point $x_t$ that was queried at time $t$. We know $x_t^\top A_t^{-1} x_t > T^{-\kappa}$. First note that any points after time $t$ do not affect the query condition at time $t$, so we only focus on deletions of $x_j$ where $j < t$.

First consider the case that $x_j$ was not queried. Then $x_t^\top A_{t \smallsetminus x_j}^{-1} x_t = x_t^\top A_t^{-1} x_t > T^{-\kappa}$. Otherwise, in the case that $x_j$ was queried

$$
\begin{aligned}
x_t^\top A_{t \smallsetminus x_j}^{-1} x_t &= x_t^\top (A_t - x_j x_j^\top)^{-1} x_t \\
&= x_t^\top A_t^{-1} x_t + \left( \frac{x_t^\top A_t^{-1} x_j x_j^\top A_t^{-1} x_t}{1 - x_j^\top A_t^{-1} x_j} \right) \\
&= x_t^\top A_t^{-1} x_t + \frac{(x_t^\top A_t^{-1} x_j)^2}{1 - x_j^\top A_t^{-1} x_j} \\
&\geq x_t^\top A_t^{-1} x_t \\
&\geq T^{-\kappa}
\end{aligned}
$$

where the second to last line follows because the second term is always positive. Thus, $x_t$ remains queried after deletion. $\qquad\square$

**Theorem 3.** *Let $\mathfrak{C}(S)$ denote the core set of the* BBQSAMPLER *on sample $S$. [Algorithm 2](#) is an exact system-aware-unlearning algorithm ([3](#)) with $S' = \mathfrak{C}(S) \smallsetminus U$.*

**Proof.** Define $S' = \mathfrak{C}(S) \smallsetminus U$. Clearly, $S' \cap U = \varnothing$. The core set of the BBQSAMPLER is exactly the set of points that it queries. Thus, applying Lemma [1](#), we know $\mathfrak{C}(\mathfrak{C}(S) \smallsetminus U) = \mathfrak{C}(S) \smallsetminus U$. $A(S', \varnothing)$ returns an ERM over $\mathfrak{C}(\mathfrak{C}(S) \smallsetminus U)$ which is exactly $\mathfrak{C}(S) \smallsetminus U$ and stores that ERM and the set $\mathfrak{C}(S) \smallsetminus U$. To process the deletion of $U$, $A(S, U)$ returns an ERM over $\mathfrak{C}(S) \smallsetminus U$ and stores that ERM and the set $\mathfrak{C}(S) \smallsetminus U$. Thus, $\mathsf{I}(S', \varnothing) = \mathsf{I}(S, U)$ and $A(S', \varnothing) = A(S, U)$. $\qquad\square$

**Theorem 4.** *With probability at least $1 - \delta$, the excess risk of the predictor $\tilde{w}$ in [Algorithm 2](#) is bounded by*

$$
\mathsf{ExcessRisk}(\widetilde{w}) = O\left( \min_\varepsilon \left\{ \frac{T_\varepsilon}{T} + \frac{dT^\kappa \log T}{T} + \frac{\log(1/\delta)}{T} \right\} \right),
$$

*even after unlearning*

$$
K = O\left( \frac{\bar{\varepsilon}^2 \cdot T^\kappa}{d \log T} \right)
$$

*many core-set deletions, where $T_\varepsilon = \sum_{t=1}^T \mathbf{1}\{|\mathbf{u}^\top x_t| \leq \varepsilon\}$, and $\bar{\varepsilon}$ denotes the minimizing $\varepsilon$ in the excess risk bound above. Furthermore, the memory required by [Algorithm 2](#) is determined by the number of core set points which is bounded by $|\mathfrak{C}(S)| \leq O(dT^\kappa \log T)$.*

**Proof.** The bound on the query complexity of the BBQ sampler before unlearning is given by Theorem [10](#) using standard analysis for selective sampling algorithms.

Now for bounding the excess risk. First let's set all of the $T_{\bar\varepsilon}$ margin points aside. Let $w_T$ be the last predictor from the BBQSAMPLER

First we argue that before deletion, $w_T^\top x$ and $\mathbf{u}^\top x$ agree on the sign of all unqueried points $x$ (outside of the $T_{\bar\varepsilon}$ margin points). These unqueried points $x$ have a margin of $\bar\varepsilon$ with respect to $w^*$, which means $|\mathbf{u}^\top x| > \bar\varepsilon$. We also have

$$
\begin{aligned}
|w_T^\top x - \mathbf{u}^\top x| &= \|w_T - \mathbf{u}\|_{A_T} \cdot \|x\|_{A_T} \\
&\leq \|w_T - \mathbf{u}\|_{A_T} \cdot \|x\|_{A_t} \qquad \text{(using the monotonicity of the query condition)} \\
&\leq \sqrt{d \log T} \cdot T^{-\kappa}
\end{aligned}
$$
$$
\text{(using Agarwal (2013) Proposition 1 and the query condition bound)}
$$

$$\leq \frac{\bar{\varepsilon}}{2} \qquad \text{(for sufficiently large T)}$$

Thus, $\text{sign}(w_T^\top x) = \text{sign}(\mathbf{u}^\top x)$, so $w_T$ correctly classifies all of the unqueried points. Furthermore, all of the unqueried points $x$ have a margin of $\frac{\bar{\varepsilon}}{2}$ with respect to $w_T$.

Thus, in order to ensure that $w_T$ and $w_{T \setminus U}$ after $|U| = K$ deletions continue to agree on the classification of all unqueried points, we need to ensure that $|w_T^\top x - w_{T \setminus U}^\top x| = \Delta < \frac{\bar{\varepsilon}}{2}$. Using the upper bound on $\Delta$ derived using a stability analysis in Theorem 8 , we get the following deletion capacity on queried points

$$\Delta \leq 2\sqrt{e(K+1)} \cdot T^{-\kappa/2} \cdot \sqrt{d \log T} \leq \frac{\bar{\varepsilon}}{2} \qquad \text{(Theorem 8)}$$

$$e(K+1) \cdot T^{-\kappa} \cdot d \log T \leq \frac{\bar{\varepsilon}^2}{16}$$

$$K + 1 \leq \frac{\bar{\varepsilon}^2 \cdot T^\kappa}{16e \cdot d \log T}$$

$$K \leq \frac{\bar{\varepsilon}^2 \cdot T^\kappa}{16e \cdot d \log T} - 1$$

$$K \leq O\left(\frac{\bar{\varepsilon}^2 \cdot T^\kappa}{d \log T}\right)$$

For up to $K$ deletions on queried points, $w_T$ and $w_{T \setminus U}$ are guaranteed to agree on the classification of all unqueried points. Thus after unlearning, $w_{T \setminus U}$ correctly classifies all of the unqueried points. We have no regret guarantees for $w_{T \setminus U}$ on the queried points and the $T_{\bar{\varepsilon}}$ margin points; therefore, we assume that we suffer full classification loss on these points. Thus, the empirical loss of $w_{T \setminus U}$ after unlearning is at most $O(T_{\bar{\varepsilon}} + dT^\kappa \ln T)$.

This can be converted to an excess risk bound using standard techniques from Bousquet et al. (2004). For a model class with VC dimension $h$ and a predictor $\hat{f}$ with empirical loss $\hat{R}$ on a sample of size $T$, we have that the excess risk is

$$\mathbb{E}[\mathbf{1}\{\hat{f}(x) \neq y\} - \mathbf{1}\{f^*(x) \neq y\}] \leq \frac{3\hat{R}}{T} + \frac{6h \log n + 6 \log(4/\delta)}{T}$$

from Bousquet et al. (2004). Plugging in the empirical loss of $O(T_{\bar{\varepsilon}} + dT^\kappa \ln T)$ for $w_{T \setminus U}$ and VC dimension $h = d + 1$, we have

$$\mathbb{E}_{(x,y)\sim\mathcal{P}}[\mathbf{1}\{\text{sign}(\tilde{w}^\top x) \neq y\} - \mathbf{1}\{\text{sign}(\mathbf{u}^\top x) \neq y\}] \leq O\left(\frac{T_{\bar{\varepsilon}}}{T} + \frac{dT^\kappa \log T}{T} + \frac{\log(1/\delta)}{T}\right)$$

with probability at least $1 - \delta$. $\qquad \square$

**Theorem 5.** *Consider any core-set algorithm $A$. Let $\pi$ denote denote a uniformly random permutation of the samples in $S$, and let $\sigma$ be a sequence of deletion requests samples from $\mu$, without replacement. Further, let $K_{\text{CSD}}$ denote the number of core set deletions within the first $K_{\text{TOTAL}}$ deletion requests, then for any $K \geq 1$,*

$$\text{Pr}_{S,\pi,\sigma}(K_{\text{CSD}} > K) \leq \frac{1}{K}\mathbb{E}_S\left[\sum_{t=1}^T \mathbb{E}_\pi[\mathbf{1}\{x_t \in \mathfrak{C}_A(\pi(S))\}] \cdot \sum_{k=1}^{K_{\text{TOTAL}}} \mathbb{E}_\sigma[\mathbf{1}\{x_t = x_{\sigma_k}\}]\right].$$

*where $\mathfrak{C}_A(\pi(S))$ denotes the coreset resulting from running $A$ on the permuted dataset $\pi(S)$. Instantiating the above bound for Algorithm 2 implies that*

$$\text{Pr}_{S,\pi,\sigma}(K_{\text{CSD}} > K) \leq \frac{K_{\text{TOTAL}} \cdot T^\kappa}{K}\mathbb{E}_S[\mathbb{E}_{x\sim\mu}[x^\top \overline{M} x]]$$

*where $\overline{M} := \mathbb{E}_\pi[\frac{1}{T} \sum_{s=1}^T A_{s-1}^{-1}]$ and $\kappa \in (0,1)$ denotes the parameter for Algorithm 2.*

**Proof.**

$$\text{Pr}_{S,\pi,\sigma}(K_{\text{CSD}} > K) \leq \frac{1}{K}\mathbb{E}[K_{\text{CSD}}] \qquad \text{(Markov's Inequality)}$$

$$= \frac{1}{K} \mathbb{E}_{S,\pi,\sigma} \left[ \sum_{t=1}^{T} \mathbf{1}\{x_t \in C_\pi\} \cdot \sum_{k=1}^{K_{\text{TOTAL}}} \mathbf{1}\{x_t = x_{\sigma_k}\} \right]$$

$$(C_\pi \text{ is the resulting core set after executing on } \pi(S))$$

$$= \frac{1}{K} \mathbb{E}_{S} \left[ \sum_{t=1}^{T} \mathbb{E}_\pi [\mathbf{1}\{x_t \in C_\pi\}] \cdot \sum_{k=1}^{K_{\text{TOTAL}}} \mathbb{E}_\sigma [\mathbf{1}\{x_t = x_{\sigma_k}\}] \right]$$

$$= \frac{1}{K} \mathbb{E}_{S,\sigma} \left[ \sum_{t=1}^{T} \mathbb{E}_\pi [\mathbf{1}\{x_t \in C_\pi\}] \cdot \sum_{k=1}^{K_{\text{TOTAL}}} \mathbf{1}\{x_t = x_{\sigma_k}\} \right]$$

$$= \frac{1}{K} \mathbb{E}_{S,\sigma} \left[ \sum_{k=1}^{K_{\text{TOTAL}}} \mathbb{E}_\pi [\mathbf{1}\{x_{\sigma_k} \in C_\pi\}] \right]$$

This proves the first half of the theorem.

Recall the following theorem from Ben-Hamou et al. (2018).

**Theorem 7.** *Let $X$ be the cumulative value of sequence of length $n \leq N$ drawn from $\Omega$ without replacement,*

$$X = \nu(\mathbf{I}_1) + \cdots + \nu(\mathbf{I}_n),$$

*and let $Y$ be the cumulative value of sequence of length $n \leq N$ drawn from $\Omega$ with replacement,*

$$X = \nu(\mathbf{J}_1) + \cdots + \nu(\mathbf{J}_n).$$

*If the value function $\nu$ and the weight vector $W$ follow the property that*

$$\omega(i) > \omega(j) \implies \nu(i) \geq \nu(j),$$

*Then*

$$\mathbb{E}[X] \leq \mathbb{E}[Y]$$

Consider the case when $\nu(x) = \mathbb{E}_\pi [\mathbf{1}\{x \in C_\pi\}]$ and the deletion distribution $\mu$ satisfies $\mu(x) > \mu(x') \implies \nu(x) \geq \nu(x')$. This is exactly the worst case in terms of deletion capacity: points that have a high probability of being included in the core set are exactly the points that have a high probability of being deleted.

In this case, we can apply Theorem 7 to get

$$\Pr_{S,\pi,\sigma}(X > K) \leq \frac{1}{K} \mathbb{E}_{S,\sigma} \left[ \sum_{k=1}^{K_{\text{TOTAL}}} \mathbb{E}_\pi [\mathbf{1}\{x_{\sigma_k} \in C_\pi\}] \right]$$

$$\leq \frac{1}{K} \mathbb{E}_{S} \left[ \mathbb{E}_{x \sim \mu} \sum_{k=1}^{K_{\text{TOTAL}}} \left[ \mathbb{E}_\pi [\mathbf{1}\{x \in C_\pi\}] \right] \right]$$

$$(\text{where } x \text{ is sampled without replacement from } W)$$

$$\leq \frac{K_{\text{TOTAL}}}{K} \mathbb{E}_{S} \left[ \mathbb{E}_{x \sim \mu} \left[ \mathbb{E}_\pi [\mathbf{1}\{x \in C_\pi\}] \right] \right]$$

$$\leq \frac{K_{\text{TOTAL}}}{K} \mathbb{E}_{S} \left[ \mathbb{E}_{x \sim \mu} \left[ \frac{T^\kappa}{T} \sum_{s=1}^{T} x^\top \mathbb{E}_\pi [A_{s-1}^{-1}] x \right] \right]$$

$$(\text{plugging in upper bound for } \mathbb{E}_\pi [\mathbf{1}\{x \in C_\pi\}])$$

$$\leq \frac{K_{\text{TOTAL}} \cdot T^\kappa}{K \cdot T} \mathbb{E}_{S} \left[ \mathbb{E}_{x \sim \mu} \left[ \sum_{s=1}^{T} x^\top \mathbb{E}_\pi [A_{s-1}^{-1}] x \right] \right]$$

$$\leq \frac{K_{\text{TOTAL}} \cdot T^\kappa}{K} \mathbb{E}_{S} \left[ \mathbb{E}_{x \sim \mu} \left[ x^\top \mathbb{E}_\pi \left[ \frac{1}{T} \sum_{s=1}^{T} A_{s-1}^{-1} \right] x \right] \right]$$

$$\leq \frac{K_{\text{TOTAL}} \cdot T^\kappa}{K} \mathbb{E}_{S} [\mathbb{E}_{x \sim \mu} [x^\top \overline{M} x]]$$

where $\overline{M} = \mathbb{E}_\pi [\frac{1}{T} \sum_{s=1}^{T} A_{s-1}^{-1}]$ for a given sample $S$. $\qquad \square$

**Lemma 3.** *Let the deletion distribution $\mu$ be the uniform distribution. Working out the bound from [Theorem 5](#), we have*

$$\Pr_{S,\pi,\sigma}(K_{\text{CSD}} > K) \leq \frac{K_{\text{TOTAL}} \cdot T^{\kappa} \cdot d \log T}{K \cdot T}.$$

*We can process a total of*

$$K_{\text{TOTAL}} = \frac{c \cdot K \cdot T}{d T^{\kappa} \log T}$$

*deletions such that the probability that we exhaust the core set deletion capacity of $K$ is at most $c$.*

**Proof.**

$$\mathbb{E}_S[\mathbb{E}_{x \sim \text{unif}}[x^\top \overline{M} x]] = \mathbb{E}_S\left[\frac{1}{T} \sum_{t=1}^{T} x_t^\top \overline{M} x_t\right]$$

$$\leq \frac{d \log T}{T} \qquad\qquad (\sum_{t=1}^{T} x_t A_{t-1}^{-1} x_t \leq d \log T)$$

Plugging this into [Theorem 5](#) and solving for $K_{\text{TOTAL}}$ completes the proof of the lemma. $\qquad\square$

### A.6 AUXILLARY RESULTS

**Theorem 8.** *Let $w_T$ be the final predictor after running the BBQSAMPLER from Algorithm [2](#) with $\lambda = K$. Let $D$ be a sequence of deletions of length $K$. Let $w_{T \smallsetminus U}$ be the predictor after the sequence of $D$ deletions have been applied. Let $x$ be an unqueried point. Then we have*

$$\Delta = w_{T \smallsetminus U}^\top x - w_T^\top x - \leq 2\sqrt{e(K+1)} \cdot T^{-\kappa/2} \cdot \sqrt{d \log T}$$
$$= O\left(\sqrt{K} \cdot T^{-\kappa/2} \cdot \sqrt{d \log T}\right)$$

**Proof.** Let $D_i$ be the set of the first $i$ deletions.

$$\Delta = w_{T \smallsetminus U}^\top x - w_T^\top x$$

$$= \sum_{i=1}^{K} (w_{T \smallsetminus U_i}^\top x - w_{T \smallsetminus U_{i-1}}^\top x)$$

$$= \sum_{i=1}^{K} \frac{2\sqrt{e(K+1)}}{K} \cdot T^{-\kappa/2} \cdot \sqrt{d \log T} \qquad\qquad \text{(applying Theorem [9](#))}$$

$$\leq \frac{2K\sqrt{e(K+1)}}{K} \cdot T^{-\kappa/2} \cdot \sqrt{d \log T}$$

$$\leq 2\sqrt{e(K+1)} \cdot T^{-\kappa/2} \cdot \sqrt{d \log T}$$

$$\square$$

**Theorem 9.** *Let $\lambda = K$ be the regularization parameter. Consider a set of $D$ deletions where $|U| < K$. Let $w_{T \smallsetminus U}$ be the predictor after the set of $D$ deletions have been applied and let $w_{T \smallsetminus (D \cup x_i)}$ be the predictor after the set of $D$ deletions have been applied along with an additional deletion of $x_i$. Let $x$ be an unqueried point. Then we have*

$$\Delta = w_{T \smallsetminus (D \cup x_i)}^\top x - w_{T \smallsetminus U}^\top x \leq \frac{2\sqrt{e(K+1)}}{K} \cdot T^{-\kappa/2} \cdot \sqrt{d \log T}$$

*for $\lambda = 1$.*

**Proof.** Let $A_{T \setminus U} = A_T - \sum_{j \in U} x_j x_j^\top$ and $b_{T \setminus U} = b_T - \sum_{j \in U} y_j x_j$

$$
\begin{aligned}
\Delta &= w_{T \setminus (D \cup x_i)}^\top x - w_{T \setminus U}^\top x \\
&= (b_{T \setminus U} - y_i x_i)^\top (A_{T \setminus U} - x_i x_i^\top)^{-1} x - b_{T \setminus U}^\top A_{T \setminus U}^{-1} x \\
&= b_{T \setminus U}^\top (A_{T \setminus U} - x_i x_i^\top)^{-1} x - y_i x_i^\top (A_{T \setminus U} - x_i x_i^\top)^{-1} x - b_{T \setminus U}^\top A_{T \setminus U}^{-1} x \\
&= b_{T \setminus U}^\top A_{T \setminus U}^{-1} x + \left( \frac{b_{T \setminus U}^\top A_{T \setminus U}^{-1} x_i x_i^\top A_{T \setminus U}^{-1} x}{1 - x_i^\top A_{T \setminus U}^{-1} x_i} \right) - y_i x_i^\top A_{T \setminus U}^{-1} x - y_i \left( \frac{x_i^\top A_{T \setminus U}^{-1} x_i x_i^\top A_{T \setminus U}^{-1} x}{1 - x_i^\top A_{T \setminus U}^{-1} x_i} \right) - b_{T \setminus U}^\top A_{T \setminus U}^{-1} x
\end{aligned}
$$
$$\text{(Sherman-Morrison)}$$

$$
\begin{aligned}
&= \left( \frac{b_{T \setminus U}^\top A_{T \setminus U}^{-1} x_i x_i^\top A_{T \setminus U}^{-1} x}{1 - x_i^\top A_{T \setminus U}^{-1} x_i} \right) - y_i x_i^\top A_{T \setminus U}^{-1} x - y_i \left( \frac{x_i^\top A_{T \setminus U}^{-1} x_i x_i^\top A_{T \setminus U}^{-1} x}{1 - x_i^\top A_{T \setminus U}^{-1} x_i} \right) \\
&= \left( \frac{w_{T \setminus U}^\top x_i x_i^\top A_{T \setminus U}^{-1} x}{1 - x_i^\top A_{T \setminus U}^{-1} x_i} \right) - y_i x_i^\top A_{T \setminus U}^{-1} x - y_i \left( \frac{x_i^\top A_{T \setminus U}^{-1} x_i x_i^\top A_{T \setminus U}^{-1} x}{1 - x_i^\top A_{T \setminus U}^{-1} x_i} \right) \\
&= \left( \frac{w_{T \setminus U}^\top x_i x_i^\top A_{T \setminus U}^{-1} x}{1 - \frac{1}{\lambda+1}} \right) - y_i x_i^\top A_{T \setminus U}^{-1} x - y_i \left( \frac{x_i^\top A_{T \setminus U}^{-1} x}{(\lambda+1)(1 - \frac{1}{\lambda+1})} \right)
\end{aligned}
$$
$$(x_i^\top A_{T \setminus U}^{-1} x_i \le \tfrac{1}{\lambda+1} \text{ from Lemma } 2)$$

$$
\begin{aligned}
&= \frac{1}{1 - \frac{1}{\lambda+1}} \cdot w_{T \setminus U}^\top x_i \cdot x_i^\top A_{T \setminus U}^{-1} x - y_i x_i^\top A_{T \setminus U}^{-1} x - \frac{1}{\lambda} y_i x_i^\top A_{T \setminus U}^{-1} x \\
&= \frac{1}{1 - \frac{1}{\lambda+1}} \cdot w_{T \setminus U}^\top x_i \cdot x_i^\top A_{T \setminus U}^{-1} x - \left( 1 + \frac{1}{\lambda} \right) y_i x_i^\top A_{T \setminus U}^{-1} x \\
&= \left( 1 + \frac{1}{\lambda} \right) x_i^\top A_{T \setminus U}^{-1} x \cdot (w_{T \setminus U}^\top x_i - y_i) \\
&\le \left( 1 + \frac{1}{\lambda} \right) (w_{T \setminus U}^\top x_i - y_i) \cdot \sqrt{x_i^\top A_{T \setminus U}^{-1} x_i \cdot x^\top A_{T \setminus U}^{-1} x} &&\text{(applying Lemma } 3) \\
&\le \left( \frac{\lambda+1}{\lambda} \right) (w_{T \setminus U}^\top x_i - y_i) \cdot \sqrt{\frac{e}{\lambda+1} \cdot T^{-\kappa}} &&\text{(applying Lemma } 2 \text{ and Corollary } 1) \\
&= \frac{\sqrt{e(\lambda+1)}}{\lambda} \cdot T^{-\kappa/2} (w_{T \setminus U}^\top x_i - y_i) \\
&= \frac{\sqrt{e(\lambda+1)}}{\lambda} \cdot T^{-\kappa/2} (w_{T \setminus U}^\top x_i - \mathbf{u}^\top x_i + \zeta_i) \\
&= \frac{\sqrt{e(\lambda+1)}}{\lambda} \cdot T^{-\kappa} (w_{T \setminus U}^\top x_i - \mathbf{u}^\top x_i) + \frac{\sqrt{e(\lambda+1)}}{\lambda} \cdot \zeta_i \cdot T^{-\kappa/2} \\
&\le \frac{\sqrt{e(\lambda+1)}}{\lambda} \cdot T^{-\kappa/2} \|w_{T \setminus U} - \mathbf{u}\|_{A_{T \setminus U}} \|x_i\|_{A_{T \setminus U}^{-1}} + \frac{\sqrt{e(\lambda+1)}}{\lambda} \cdot \zeta_i \cdot T^{-\kappa/2} \\
&\le \frac{\sqrt{e(\lambda+1)}}{\lambda} \cdot T^{-\kappa/2} \|w_{T \setminus U} - \mathbf{u}\|_{A_{T \setminus U}} \|x_i\|_{A_{T \setminus U}^{-1}} + \frac{\sqrt{e(\lambda+1)}}{\lambda} \cdot T^{-\kappa/2} &&(\zeta_i < 1) \\
&\le \frac{\sqrt{e(\lambda+1)}}{\lambda} \cdot T^{-\kappa/2} \cdot \sqrt{d \log(T - K)} + \frac{\sqrt{e(\lambda+1)}}{\lambda} \cdot T^{-\kappa/2}
\end{aligned}
$$
$$\text{(Proposition 1 from Agarwal (2013))}$$

$$
\le \frac{2\sqrt{e(\lambda+1)}}{\lambda} \cdot T^{-\kappa/2} \cdot \sqrt{d \log T}
$$

$$\square$$

**Lemma 1.** *Let $\lambda$ be the regularization parameter. Let $U$ be a set of deletions such that $|U| = K$. Let $A_{T \setminus U}$ denote $A_T - \sum_{x_j \in U} x_j x_j^\top$. Then we have*

$$
x^\top A_{T \setminus U}^{-1} x \le \sum_{i=0}^{K} \frac{\binom{K}{i}}{\lambda^i} \cdot T^{-\kappa}.
$$

We prove the claim using induction. Assume that $x^\top A_{T \setminus U}^{-1} x \leq \sum_{i=0}^{K} \frac{\binom{K}{i}}{\lambda^i} \cdot T^{-\kappa}$ (induction hypothesis). Consider an additional deletion and the effect on the query condition, $x^\top (A_{T \setminus U} - x_i x_i)^{-1} x$

$$
\begin{aligned}
x^\top (A_{T \setminus U} - x_i x_i^\top)^{-1} x &= x^\top A_{T \setminus U}^{-1} x + \left( \frac{x^\top A_{T \setminus U}^{-1} x_i x_i^\top A_{T \setminus U}^{-1} x}{1 - x_i^\top A_{T \setminus U}^{-1} x_i} \right) \\
&\leq x^\top A_{T \setminus U}^{-1} x + \left( \frac{(x_i^\top A_{T \setminus U}^{-1} x)^2}{\left(1 - \frac{1}{\lambda+1}\right)} \right) \quad (x_i^\top A_{T \setminus (D \setminus x_i)}^{-1} x_j \leq \frac{1}{\lambda+1} \text{ from Lemma 2}) \\
&\leq \sum_{i=0}^{K} \frac{\binom{K}{i}}{\lambda^i} \cdot T^{-\kappa} + \left( \frac{(x_i^\top A_{T \setminus U}^{-1} x)^2}{\left(1 - \frac{1}{\lambda+1}\right)} \right) \quad (x^\top A_{T \setminus U}^{-1} x \leq \sum_{i=0}^{K} \frac{\binom{K}{i}}{\lambda^i} \cdot T^{-\kappa} \text{ from IH}) \\
&\leq \sum_{i=0}^{K} \frac{\binom{K}{i}}{\lambda^i} \cdot T^{-\kappa} + \left( \frac{x_i^\top A_{T \setminus U}^{-1} x_i \cdot x_i^\top A_{T \setminus U}^{-1} x}{\left(1 - \frac{1}{\lambda+1}\right)} \right) \quad \text{(Lemma 3)} \\
&\leq \sum_{i=0}^{K} \frac{\binom{K}{i}}{\lambda^i} \cdot T^{-\kappa} + \left( \frac{\sum_{i=0}^{K} \frac{\binom{K}{i}}{\lambda^i} \cdot T^{-\kappa}}{(\lambda+1)\left(1 - \frac{1}{\lambda+1}\right)} \right) \quad \text{(using IH and Lemma 2)} \\
&\leq \sum_{i=0}^{K} \frac{\binom{K}{i}}{\lambda^i} \cdot T^{-\kappa} + \sum_{i=0}^{K} \frac{\binom{K}{i}}{\lambda^{i+1}} \cdot T^{-\kappa} \\
&\leq \sum_{i=0}^{K} \frac{\binom{K}{i}}{\lambda^i} \cdot T^{-\kappa} + \sum_{i=1}^{K+1} \frac{\binom{K}{i-1}}{\lambda^i} \cdot T^{-\kappa} \\
&\leq \frac{\binom{K}{0}}{\lambda^0} \cdot T^{-\kappa} + \sum_{i=1}^{K} \frac{\binom{K}{i}}{\lambda^i} \cdot T^{-\kappa} + \sum_{i=1}^{K} \frac{\binom{K}{i-1}}{\lambda^i} \cdot T^{-\kappa} + \frac{\binom{K}{K}}{\lambda^{K+1}} \cdot T^{-\kappa} \\
&\leq \frac{\binom{K}{0}}{\lambda^0} \cdot T^{-\kappa} + \sum_{i=1}^{K} \frac{\binom{K+1}{i}}{\lambda^i} \cdot T^{-\kappa} + \frac{\binom{K}{K}}{\lambda^{K+1}} \cdot T^{-\kappa} \quad \text{(Pascal's Identity)} \\
&\leq \frac{\binom{K}{0}}{\lambda^0} \cdot T^{-\kappa} + \sum_{i=1}^{K} \frac{\binom{K+1}{i}}{\lambda^i} \cdot T^{-\kappa} + \frac{\binom{K+1}{K+1}}{\lambda^{K+1}} \cdot T^{-\kappa} \\
&\leq \sum_{i=0}^{K+1} \frac{\binom{K+1}{i}}{\lambda^i} \cdot T^{-\kappa}
\end{aligned}
$$

**Corollary 1.** *Let $\lambda = K$ be the regularization parameter. Let $U$ be a set of deletions such that $|U| < K$, then $x^\top A_{T \setminus U}^{-1} x \leq e \cdot T^{-\kappa}$*

**Proof.** Note that:

$$
\begin{aligned}
x^\top A_{T \setminus U}^{-1} x &\leq \sum_{i=0}^{|U|} \frac{\binom{|U|}{i}}{\lambda^i} \cdot T^{-\kappa} \quad \text{(Lemma 1)} \\
&\leq \sum_{i=0}^{K} \frac{\binom{K}{i}}{\lambda^i} \cdot T^{-\kappa} \\
&\leq \sum_{i=0}^{K} \frac{\binom{K}{i}}{K^i} \cdot T^{-\kappa} \\
&= \left(1 + \frac{1}{K}\right)^K \cdot T^{-\kappa} \\
&\leq e \cdot T^{-\kappa}
\end{aligned}
$$

$\square$

**Lemma 2.** $x_i^\top A_S^{-1} x_i \leq \frac{1}{\lambda+1}$ *for any set of $S$ points such that $x_i \in S$, where $A_S = I + \sum_{x_t \in S} x_t x_t^\top$*

**Proof.** We want to consider the $x_i$ that maximizes $x_i^\top A_S^{-1} x_i$. Let $A_{S \setminus i} = I + \sum_{x_t \in S \setminus \{x_i\}} x_t x_t^\top$. Then we want to maximize the following

$$
x_i^\top A_S^{-1} x_i = x_i^\top (A_{S \setminus i} + x_i x_i^\top)^{-1} x_i
$$

$$= x_i^\top A_{S \setminus i}^{-1} x_i - \frac{x_i^\top A_{S \setminus i}^{-1} x_i x_i^\top A_{S \setminus i}^{-1} x_i}{1 + x_i^\top A_{S \setminus i}^{-1} x_i}$$

Let $a = x_i^\top A_{S \setminus i}^{-1} x_i$.

$$x_i^\top A_S^{-1} x_i = a - \frac{a^2}{1 + a} = \frac{a}{1 + a} = 1 - \frac{1}{1 + a}$$

We want to maximize the above expression where $0 \le a \le \frac{1}{\lambda}$ (since $0 \le x_i^\top A_{S \setminus i}^{-1} x_i \le \frac{1}{\lambda}$). The expression is maximized when $a = \frac{1}{\lambda}$. Thus, $x_i^\top A_{T-1}^{-1} x_i \le \frac{1}{\lambda(1 + \frac{1}{\lambda})} = \frac{1}{\lambda + 1}$. $\qquad \square$

**Lemma 3.** $(x_i^\top A_S^{-1} x)^2 \le x_i^\top A_S^{-1} x_i \cdot x^\top A_S^{-1} x$ *for any set of $S$ points such that $x_i \in S$, where $A_S = \lambda I + \sum_{x_t \in S} x_t x_t^\top$*

**Proof.** We can decompose the terms as $A_S^{-1} = \sum_{i=1}^d \lambda_i u_i u_i^\top$, $x_i = \sum_{i=1}^d \alpha_i u_i$, and $x = \sum_{i=1}^d \beta_i u_i$. Using these decompositions, we compute the following two terms

$$(x_i^\top A_T^{-1} x)^2 = \left( \left( \sum_{i=1}^d \alpha_i u_i^\top \right) \left( \sum_{i=1}^d \lambda_i u_i u_i^\top \right) \left( \sum_{i=1}^d \beta_i u_i \right) \right)^2$$

$$= \left( \sum_{i=1}^d \lambda_i \alpha_i \beta_i \right)^2$$

$$x_i^\top A_T^{-1} x_i \cdot x^\top A_T^{-1} x = \left( \sum_{i=1}^d \alpha_i u_i^\top \right) \left( \sum_{i=1}^d \lambda_i u_i u_i^\top \right) \left( \sum_{i=1}^d \alpha_i u_i \right) \left( \sum_{i=1}^d \beta_i u_i^\top \right) \left( \sum_{i=1}^d \lambda_i u_i u_i^\top \right) \left( \sum_{i=1}^d \beta_i u_i \right)$$

$$= \left( \sum_{i=1}^d \lambda_i \alpha_i^2 \right) \left( \sum_{i=1}^d \lambda_i \beta_i^2 \right)$$

From Jensen's inequality (or Cauchy-Schwarz inequality), we know that $(\sum_{i=1}^d p_i x_i)^2 \le \sum_{i=1}^d p_i x_i^2$ where $p_i > 0$ for all $i$ and $\sum_{i=1}^d p_i = 1$ since $f(x) = x^2$ is convex. Let $p_i = \lambda_i \alpha_i^2 / (\sum_{j=1}^d \lambda_j \alpha_j^2)$ (note that all $\lambda_i$'s $> 0$) and let $x_i = \beta_i / \alpha_i$. This gives us

$$\frac{\left( \sum_{i=1}^d \lambda_i \alpha_i \beta_i \right)^2}{\left( \sum_{i=1}^d \lambda_i \alpha_i^2 \right)^2} \le \frac{\sum_{i=1}^d \lambda_i \alpha_i^2 \cdot \frac{\beta_i^2}{\alpha_i^2}}{\sum_{i=1}^d \lambda_i \alpha_i^2}$$

$$\left( \sum_{i=1}^d \lambda_i \alpha_i \beta_i \right)^2 \le \left( \sum_{i=1}^d \lambda_i \alpha_i^2 \right) \left( \sum_{i=1}^d \lambda_i \beta_i^2 \right)$$

This directly implies that $(x_i^\top A_T^{-1} x)^2 \le x^\top A_T^{-1} x \cdot x_i^\top A_T^{-1} x_i$. $\qquad \square$

**Theorem 10.** *Let $\lambda = K \le T$ be the regularization parameter and $0 < \kappa < 1$ be the sampling parameter of the* BBQSAMPLER. *Then we have the following regret and query complexity bounds on the* BBQSAMPLER

$$R_T = \min_\varepsilon \varepsilon T_\varepsilon + O\left( \frac{1}{\varepsilon} \left( K + d \log T + \log \frac{T}{\delta} \right) + \frac{1}{\varepsilon^{2/\kappa}} \right)$$

$$N_T = O(dT^\kappa \log T)$$

**Proof.** Let $\Delta_t = \mathbf{u}^\top x_t$ and $\hat{\Delta}_t = w_t^\top x$. We decompose the regret as follows

$$R_T \le \varepsilon T_\varepsilon + \sum_{t=1}^T \bar{Z}_t \mathbf{1}\{\Delta_t \hat{\Delta}_t < 0, \Delta_t^2 > \varepsilon^2\} + \sum_{t=1}^T Z_t \mathbf{1}\{\Delta_t \hat{\Delta}_t < 0, \Delta_t^2 > \varepsilon^2\} |\Delta_t|$$

$$= \varepsilon T_\varepsilon + U_\varepsilon + Q_\varepsilon \qquad \text{(regret decomposition from Dekel et al. (2012) Lemma 3)}$$

We define an additional term

$$\hat{\Delta}'_t = \begin{cases} \text{sign}(\hat{\Delta}_t) & \text{if } |\hat{\Delta}_t| > 1 \\ \hat{\Delta}_t & \text{otherwise} \end{cases}$$

$$Q_\varepsilon \le \frac{1}{\varepsilon} \sum_{t=1}^T Z_t \mathbf{1}\{\hat{\Delta}_t \Delta_t < 0\} \Delta_t^2$$

$$= \frac{1}{\varepsilon} \sum_{t=1}^T Z_t \mathbf{1}\{\hat{\Delta}'_t \Delta_t < 0\} \Delta_t^2 \qquad (\hat{\Delta}_t \text{ and } \hat{\Delta}'_t \text{ have the same sign})$$

$$\le \frac{1}{\varepsilon} \sum_{t=1}^T Z_t (\Delta_t - \hat{\Delta}_t)^2 \qquad (\hat{\Delta}'_t \Delta_t < 0 \text{ implies } \Delta_t^2 \le (\Delta_t - \hat{\Delta}'_t)^2)$$

$$\le \frac{2}{\varepsilon} \left( \sum_{t=1}^T Z_t ((\Delta_t - y)^2 - (\hat{\Delta}_t - y)^2) + 144 \log \frac{T}{\delta} \right) \qquad \text{(Dekel et al. (2012) Lemma 23 (i))}$$

$$\le \frac{4}{\varepsilon} \left( \sum_{t=1}^T Z_t \left( d_{t-1}(w^*, w_{t-1}) - d_t(w^*, w_t) + 2 \log \frac{|A_t|}{|A_{t-1}|} \right) + 144 \log \frac{T}{\delta} \right)$$
$$\text{(Dekel et al. (2012) Lemma 25 (iv) where } d_t(w^*, w) = \tfrac{1}{2}(w^* - w)^\top A_t (w^* - w))$$

$$\le \frac{4}{\varepsilon} \left( d_0(w^*, w_0) + \log |A_T| + 144 \log \frac{T}{\delta} \right)$$

$$\le \frac{2}{\varepsilon} \left( \lambda + d \log(\lambda + N_T) + 144 \log \frac{T}{\delta} \right) \qquad \text{(Dekel et al. (2012) Lemma 24 (iii))}$$

$$= O\left( \frac{1}{\varepsilon} \left( \lambda + d \log T + \log \frac{T}{\delta} \right) \right)$$

Let $r_t = x_t^\top A_t^{-1} x_t$

$$U_\varepsilon \le \sum_{t=1}^T \bar{Z}_t \, \mathbf{1}\{|\hat{\Delta}_t - \Delta_t| > \varepsilon\}$$

$$\le (2 + e) \sum_{t=1}^T \bar{Z}_t \exp\left( -\frac{\varepsilon^2}{8 r_t} \right) \qquad \text{(following Cesa-Bianchi et al. (2009) Theorem 1)}$$

$$= (2 + e) \sum_{t=1}^T \bar{Z}_t \exp\left( -\frac{\varepsilon^2 T^\kappa}{8} \right) \qquad \text{(when } \bar{Z}_t = 1, r_t < T^{-\kappa} \text{ by the query condition)}$$

$$\le (2 + e) \sum_{t=1}^T \bar{Z}_t \exp\left( -\frac{\varepsilon^2 t^\kappa}{8} \right)$$

$$\le (2 + e) \lceil 1/\kappa \rceil! \left( \frac{8}{\varepsilon^2} \right)^{1/\kappa} \qquad \text{(following Cesa-Bianchi et al. (2009) Theorem 1)}$$

$$\le O\left( \frac{1}{\varepsilon^{2/\kappa}} \right)$$

Putting the above terms together completes the proof of regret.

Now for the number of queries. Let $r_t = x_t^\top A_t^{-1} x_t$. Consider the following sum

$$\sum_{t=1}^T Z_t r_t \le \sum_{t=1}^T Z_t \cdot \log \frac{|A_t|}{|A_{t-1}|} \qquad \text{(Lemma 24 from Dekel et al. (2012) where } |\cdot| \text{ is the determinant)}$$

$$= \log \frac{|A_T|}{|A_0|}$$

$$\le \log |A_T|$$

$$\le d \log(\lambda + N_T)$$

$$\le d \log(T)$$

We use the above sum to bound the number of queries

$$
\begin{aligned}
N_T &= \sum_{r_t > T^{-\kappa}} 1 \\
&\leq \sum_{r_t > T^{-\kappa}} \frac{r_t}{T^{-\kappa}} \\
&\leq T^\kappa \sum_{r_t > T^{-\kappa}} r_t \\
&\leq O(dT^\kappa \log(T)) \qquad \text{(using the sum above)}
\end{aligned}
$$

$\square$

