# OpenReview forum: "System Aware Unlearning Algorithms: Use Lesser, Forget Faster"
_ICLR.cc/2025/Conference — Submitted to ICLR 2025_

### Official Review · Reviewer_LA9z · 2024-10-29

**Soundness:** 3
**Presentation:** 2
**Contribution:** 3
**Rating:** 6
**Confidence:** 4

**Summary:**

The paper begins by discussing existing definitions of machine unlearning, highlighting their limitations through a specific case study. It then introduces a novel definition, System-Aware Unlearning, and provides a detailed explanation.
Contributions:
1.	The paper analyzes the limitations of the existing unlearning definitions and introduces the concept of System-Aware Unlearning.
2.	They propose a general-purpose unlearning algorithm utilizing data sharding under the framework of System-Aware Unlearning.
3.	Authors also propose an unlearning algorithm for linear classification using selective sampling for the special case of linear classification.

**Strengths:**

1.	The paper introduces a novel System-Aware Unlearning framework that advances the theoretical foundations of machine unlearning. This framework takes a more practical approach by considering the actual system security model and attacker capabilities.
2.	This paper includes two algorithms: a general-purpose algorithm for exact system-aware unlearning using data sharding, and an improved system-aware unlearning algorithm for the special case of linear classification.
3.	A comprehensive theoretical analysis is provided that includes deletion capacity, model accuracy, memory requirements, and computational complexity.

**Weaknesses:**

1.	The necessity of "system aware unlearning" is not well justified. A critical assumption in the Introduction states, "consider a learning algorithm that relies on only a fraction of its training dataset to generate its hypothesis and hence the ML system only stores this data. " However, the paper fails to explain why models would be trained on partial datasets, or whether the data selection process itself requires unlearning due to potential information leakage.
2.	The paper claims, " Even if an observer/attacker has access to larger public data sets that might include parts of the data the system was trained on, in such a system we could expect privacy for data that the system does not use directly for building the model to be preserved." This statement is not true, as no privacy can be preserved if the entire training dataset is exposed to attackers.
3.	The conclusion that models would "not be statistically indistinguishable from each other" on Page 3 does not hold when the model owner uses the complete training dataset S.
4.	Using only one shard's core set for prediction would likely cause significant performance degradation compared to models trained on complete datasets. The paper lacks experimental validation of this approach.
5.	The paper lacks an Evaluation section. Machine unlearning algorithms should be assessed across multiple dimensions, including unlearning efficiency, effectiveness, and model utility preservation. A comparative analysis with existing methods is also necessary.
6.	The paper lacks a Related Work section.
7.	Technical writing requires improvement, particularly in defining key concepts like ExcessRisk and parameters such as f*.

**Questions:**

Please see the weaknesses.

---

> ### Author Response · Authors · 2024-11-22
> **Response to Reviewer LA9z**
>
> We thank the reviewer for their thoughtful feedback. We have addressed writing concerns. We have uploaded an updated paper pdf which incorporates their feedback, and we address their comments below.
>
> > **The necessity of "system aware unlearning" is not well justified…**
>
> There already exist learning algorithms that by design, only rely on a small fraction of the training dataset in order to generate a hypothesis, such as selective sampling algorithms [1], which are discussed in detail in Section 4. For another concrete example, consider a hard margin SVM: only the training points on the margin dictate the decision boundary. Most of the points are not on the margin and do not affect the decision boundary. If a non-margin point is deleted, the decision boundary and hypothesis remain unchanged. Furthermore, the algorithms in the paper demonstrate that when only a small fraction of the training data is used to generate a hypothesis, we can unlearn faster and use less memory. The goal of efficient unlearning motivates the choice of relying on fewer data points. On the information leakage question, it can be shown that Given the state of the system $\textsf{I}(S, U)$, the conditional mutual information between $U$ and $A(S, U)$ is 0. We have added this discussion to the paper in Section 2 in lines 171-183.
>
> > **The paper claims, " Even if an observer/attacker has access to larger public data sets…**
>
> While making this claim, we do not assume that the entire training dataset is exposed to the attackers. We simply claim that the part of the dataset that is *not used* in training the model, and additionally not exposed to the adversary by other methods, enjoys privacy even if the final state of the system (which again does not contain these samples that were not used in training) is exposed to the attacker. We are happy to clarify further if something is still confusing.
>
> When the system is compromised, we assume that the attacker only gains knowledge of the points in the core set of the model. Now suppose that the attacker had access to some auxiliary dataset of data points, some of which could have appeared in the sample but outside of the core set. The attacker cannot identify which of these points actually appeared in the sample because the output of the model is exactly the same whether or not one of these points actually appeared in the sample of not, since points outside of the core set have no effect on the model. Thus, we maintain privacy with respect to membership in the sample even in the presence of auxiliary information.
>
> > **The conclusion that models would "not be statistically indistinguishable from each other"…**
>
> In the motivating example on page 3, the model owner only uses a small subset $C$ of $S$ and does not use the complete training dataset $S$. The motivating example is specific to the case where the model and system only use a small fraction of the training dataset $S$.
>
> > **Using only one shard's core set for prediction would likely cause…**
>
> Although Algorithm 1 incurs an initial accuracy tradeoff before deletion, as the number of deletions increases, Algorithm 1 is able to maintain much better accuracy compared to algorithms like SISA from [2] under deletion sequences which are label dependent and cause a large distribution shift.
>
> > **The paper lacks an Evaluation section. Machine unlearning algorithms should be assessed across multiple dimensions…**
>
> Our primary contribution is theoretical; we prove theoretical guarantees on model accuracy after unlearning, memory requirements, and computation time. We prove that under the assumption of a weaker adversary, we can design significantly improved memory and computationally efficient exact unlearning algorithms. Exact unlearning algorithms under the previous definition of unlearning all involve some level of retraining from scratch for each deletion and storing the entirety of the dataset, which is expensive and inefficient in memory and time. Under our new definition, we can avoid this retraining; we can design algorithms that provably beat the time to retrain and provably require significantly less memory than the entirety of the dataset, so the comparison and improvement are straightforward. In Appendix A.1, we provide some simple experiments for the linear case to validate our theoretical results. In the final version of the paper, we will bring the experiments to the main body.
>
> > **The paper lacks a Related Work section.**
>
> We discuss related unlearning definitions in Section 2, and we provide a discussion of other related work in unlearning in Appendix A.2. In the final version of the paper, we will bring the discussion of other related work to the main body.
>
> [1] https://arxiv.org/pdf/2307.04998
>
> [2] https://arxiv.org/abs/1912.03817

---

> > ### Comment · Reviewer_LA9z · 2024-11-22
> >
> > Could you please provide empirical evidence to support your claim that: “Although Algorithm 1 incurs an initial accuracy tradeoff before deletion, as the number of deletions increases, Algorithm 1 is able to maintain much better accuracy compared to algorithms like SISA from [2] under deletion sequences which are label dependent and cause a large distribution shift.”

---

> > > ### Author Response · Authors · 2024-12-03
> > > **Response to Reviewer LA9z**
> > >
> > > We provide an empirical comparison between SISA from Bourtoule et al [1] and our Algorithm 1 on two binary classification datasets: the Purchase dataset (249,215 points, dimension = 600) provided by Bourtoule et al [1] and our dataset (200,000 points, dimension = 100, margin = 0.1) used in our experiments in Appendix A.1. On these two datasets, we compared the performance of Algorithm 1 using selective sampling for sample compression and the performance of SISA under a sequence of 80,000 label-dependent deletions. We plot the resulting model accuracy over the course of these deletions for both algorithms, and we compare the memory and computation time. We summarize the results below.
> > >
> > > **Purchase dataset from Bourtoule et al [1]**
> > >
> > > Plot of model accuracy of Algorithm 1 vs SISA over the course of 80,000 label-dependent deletions: https://anonymousauthors83274.github.io/plots/
> > >
> > > | | Initial training time (secs) | Total accumulated deletion time (secs) | Percent of data stored in memory (%) |
> > > | -------- | ------- | -------- | ------- |
> > > | Algorithm 1 [ours] | 206.6 | <1 | 39.2% |
> > > | SISA [1] | 30.2 | 1174.3 | 100% |
> > >
> > > **Main Takeaways:** The initial accuracy of Algorithm 1 is slightly worse than SISA, but Algorithm 1 is able to maintain significantly better accuracy under a longer sequence of label dependent deletions compared to SISA. Algorithm 1 requires longer initial training time, but Algorithm 1 requires significantly less computation time at the time of deletion. Furthermore, Algorithm 1 requires the storage of significantly less samples.
> > >
> > > **Our dataset from Appendix A.1**
> > >
> > > Plot of model accuracy of Algorithm 1 vs SISA over the course of 80,000 label-dependent deletions: https://anonymousauthors83274.github.io/plots/
> > >
> > > | | Initial training time (secs) | Total accumulated deletion time (secs) | Percent of data stored in memory (%) |
> > > | -------- | ------- | -------- | ------- |
> > > | Algorithm 1 [ours] | 1.7 | <1 | 1.3% |
> > > | SISA [1] | 20.6 | 697.1 | 100% |
> > >
> > > **Main Takeaways:** When the dataset allows for a more favorable compression scheme, as this dataset does, Algorithm 1 is able to match the initial accuracy of SISA, despite using much less data, and Algorithm 1 is able to maintain significantly better accuracy under a longer sequence of label dependent deletions. Furthermore, Algorithm 1 requires significantly less memory and significantly less computation time, both at the time of training and the time of deletion, due to increased sample compression. When the dataset allows for significant compression, Algorithm 1 dominates SISA in accuracy, memory, and computation time.
> > >
> > > These empirical results support our theoretical analysis. We will add both the theoretical and empirical comparison of Algorithm 1 and SISA to the final version of the paper. We appreciate the reviewer's valuable feedback.
> > >
> > > [1] https://arxiv.org/abs/1912.03817

---

### Official Review · Reviewer_E7qy · 2024-11-01

**Soundness:** 3
**Presentation:** 3
**Contribution:** 2
**Rating:** 5
**Confidence:** 2

**Summary:**

(epsilon, delta) unlearning definitions require the unlearned model to be indistinguishable from retraining from scratch on the remaining data.  In this work, the authors argue that this is too strict of a definition for unlearning.  They propose a new definition for unlearning, called system aware unlearning, that takes into account the information that an attacker could recover by compromising the system.  The authors develop system aware unlearning algorithms that are efficient for a class of functions.

**Strengths:**

- “Furthermore, we believe that accounting for the information that an attacker could have access to is an interesting direction to explore in privacy, beyond unlearning.“ I fully agree, this is an interesting aspect of unlearning that prior work has neglected. Definitions that try to take this kind of information to account is valuable to the research community.

**Weaknesses:**

I fold in my questions here:

- “This presents significant challenges in the context of privacy regulations such as the
European Union’s General Data Protection Regulation (2016) (GDPR), California Consumer Privacy Act (2018) (CCPA), and Canada’s proposed Consumer Privacy Protection Act, all of which emphasize the “right to be forgotten.” As a result, there is a growing need for methods that enable the selective removal of specific training data from models that have already been trained, a process commonly referred to as machine unlearning (Cao & Yang, 2015).“ A general comment: It’s still an open problem how these pieces of legislation should apply to ML models, and whether unlearning can satisfy these legal requirements.


- “This is evidenced by a dire lack of exact/approximate unlearning algorithms beyond the simple cases of convex loss functions.“ There have been many unlearning algorithms for non-convex models, unless you mean a dire lack of unlearning algorithms that work? Although there are also exact unlearning algorithms for non-convex models such as SISA [1].

- “Our framework leverages the fact that many ML systems do not depend on the entirety of their training data equally“ Can you precisely define what you mean by this?

- Nit typo: “We then present a general-purpose algorithm for exact system-aware unlearning using data sharding for function classes that can learned using sample compression,..“

- “We also provide an improved system-aware unlearning algorithm for the special case of linear classification thus providing the first memory and time-efficient exact unlearning algorithm for linear classification.“ This confuses me. The above comment complained that there’s a lack of algorithms beyond simple convex settings and yet you provide another algorithm for the convex setting..?

- I thought that the motivating example in line 118 is bit contrived, to be statistically indistinguishable all you need to do is change the order of operations such that A() samples a set C from S and *then* removes U from this set, rather than sampling C from S\U.

- *(System-Aware-(ε, δ)-Unlearning): For any S\U there exists an S' that is distributionally indistinguishable when trained.* Does the existence of S’ imply we can find it efficiently? I think not, and if not, how should we think about instantiating this definition in practice?

- In Definition 3, taking S’=S\U recovers the original definition. I’m struggling to understand the utility of allowing S’ to be anything other than S\U. That is, I’m struggling to understand if this flexibility is useful in practice. Can you provide an illustrative example?

- “Since the attacker can only gain access to information stored by the system and used in the unlearned model, then we want to learn predictors that are dependent on a small number of samples.“ I struggle to understand this statement. For most parametric models, samples are not “stored” in the system (that is, they are stored, but through a complex learning process which is difficult/impossible to reverse engineer). Does this statement only apply to simple linear models? This idea of reducing the number of dependent samples seems to be core to the ideas underpinning section Section 3 and Section 4.

- “Algorithm 2 Unlearning algorithm for linear classification using selective sampling”. I struggle to understand if this is a real contribution to unlearning. The algorithm is simply reversing the learning process on the unlearned points through subtraction. It also applies to a very narrow case as you point out: ”This monotonicity is a unique feature of the BBQSAMPLER. Other selective sampling algorithms, such as ones from Dekel et al. (2012) or Sekhari et al. (2023), use a query condition that depends on the labels y of previously seen points. Due to the noise in these y’s, y-dependent query conditions are not monotonic; points that were queried can become unqueried. This makes it difficult and expensive to compute the core set after unlearning.“ and “ We note that since the BBQSAMPLER uses a y-independent query condition, it is suboptimal in terms of excess error before unlearning compared to algorithms from…“.

- “Why is Algorithm 2 not a valid unlearning algorithm under the prior unlearning definition
(Definition 1)? When a queried point is deleted, an unqueried point could become queried. Thus, under traditional notions of exact unlearning, during DELETIONUPDATE, not only would we have to remove the effect of U, but we would also have to add in any unqueried points that would have been queried if U never existed in S.“ I don’t follow this counterfactual argument. According to Def 1, running Algorithm 2 on S\U should be identical to running on S and then removing on U?

- Can you comment on the similarities between Algorithm 1 and SISA [1]?

[1] https://arxiv.org/abs/1912.03817

**Questions:**

See above.

---

> ### Author Response · Authors · 2024-11-22
> **Response to Reviewer E7qy Part 1**
>
> We thank the reviewer for their thoughtful feedback. We have addressed writing concerns and typos. We have uploaded an updated paper pdf which incorporates their feedback, and we address their comments below.
>
> > **“It’s still an open problem how these pieces of legislation”**
>
> We acknowledge that the laws are not yet fully formalized, and the field presents ample opportunity for both interpreting and aiding in the formalization of these laws.
>
> > **“This is evidenced by a dire lack of exact/approximate unlearning algorithms beyond…**
>
> Precisely, we mean that there is a dire lack of memory and computationally efficient unlearning algorithms for general model classes. We will clarify this in line 211 of the updated paper pdf.
>
> > **Our framework leverages the fact that many ML systems do not depend…**
>
> Consider selective sampling algorithms which have theoretical upper bounds on the number of points required to learn a good classifier [1]. In this case, only a small subset of points in the sample needs to be used to train the final classifier that still has good accuracy.
>
> > **“We also provide an improved system-aware unlearning algorithm…**
>
> Under the traditional definition of unlearning, existing lower bounds prove that any exact unlearning for linear classification must store the entire dataset in memory. Under system aware unlearning, we show that using Algorithm 1 or Algorithm 2, we can achieve exact unlearning even when storing significantly less than the entire dataset. This shows that we can achieve more efficient unlearning algorithms under system aware unlearning, and this insight could be valuable in the development of unlearning algorithms in more challenging, nonconvex settings. Furthermore, Algorithm 1 works for non-convex function classes, and we have clarified this in the paper in line 211.
>
> > **I thought that the motivating example in line 118…**
>
> The order of operations cannot be reversed. According to the previously proposed definition of unlearning, the model output after unlearning must match the counterfactual trained on $S\setminus U$. The counterfactual model can never gain access to $U$, or else you will be attempting to match a model output that can use the deleted individuals during learning, which leads to no privacy guarantees for the deleted individuals. Thus, the counterfactual model cannot select $C$ from $S$ and then remove $U$; it must select $C’$ from $S\setminus U$. On the other hand, the unlearned model must select $C$ from $S$ initially, and then remove $U$ later on, once the deletion request actually arrives.
>
> > **For any S\U there exists an S' that is distributionally indistinguishable when trained. Does the existence of S’ imply we can find it efficiently?**
>
> Sample compression and selective sampling schemes have a clear selection rule allowing algorithm designers to easily identify which points in the sample have been used to train the final classifier. Beyond sample compression, data attribution techniques [2] can be used to identify which points in the sample influence the final model output.
>
> > **In Definition 3, taking S’=S\U recovers the original definition…**
>
> Consider a classifier on $S$ such that the decision boundary is determined by some set of margin points. Now consider when some of those margin points are deleted, and the new decision boundary is decided by the remaining margin points. However, if one were to learn on $S\setminus U$, a completely different set of margin points would be chosen by the algorithm (since those original margin points never appeared) leading to a completely different classifier. The system-aware unlearning definition is not satisfied by selecting $S’=S\setminus U$, but it is satisfied by some other plausible $S’$. In the paper in lines 378-385, we have a concrete discussion about why $S’=S\setminus U$ is insufficient for the selective sampling case.
>
> > **”Since the attacker can only gain access to what is used or stored by the model…**
>
> As mentioned in this statement, we worry about samples that were either (a) stored by the model (for whatever reason), or (b) were used in training the model. In our definition, the state of the system encapsulates any samples that were stored in the system or used during training. While we agree with the reviewer that the current popular deep learning pipelines do not explicitly store the samples, we wanted a definition that goes beyond the current deep learning settings and includes other possible scenarios such as decision trees, ensembles, nearest neighbors, etc (where storing the samples makes sense).
>
> Some examples of algorithms that only rely on a small subset of the sample are sample compression-based algorithms and selective sampling and active learning methods; these schemes generalize beyond linear models [1].
>
> [1] https://arxiv.org/pdf/2307.04998
>
> [2] https://arxiv.org/pdf/2410.23232

---

> ### Author Response · Authors · 2024-11-22
> **Response to Reviewer E7qy Part 2**
>
> > **Why is Algorithm 2 not a valid unlearning algorithm under the prior unlearning definition (Definition 1)?…**
>
> Running Algorithm 2 on $S\setminus U$ is not identical to running on $S$ and then removing $U$, and this exactly motivates the reasoning behind this paper and the new definition. Algorithm 2 does not satisfy Definition 1 (prior definition of unlearning), but it still unlearns $U$. The output of Algorithm 2 after running on $S$ and then unlearning $U$ is identical to running Algorithm 2 on $\mathfrak{C}(S)\setminus U$, where $\mathfrak{C}(S)$ is the core set of $S$; this is not equivalent to running Algorithm 2 on $S\setminus U$, since C(S\setminus U) is not equal to $\mathfrak{C}(S \setminus U) \neq \mathfrak{C}(S)\setminus U$. There are points that would have appeared in $\mathfrak{C}(S \setminus U)$and contributed to the outcome of the algorithm that never appeared in the original core set $\mathfrak{C}(S)$. Thus, Algorithm 2 is not a valid unlearning algorithm under Definition 1 (the prior definition of unlearning).
>
> > **Can you comment on the similarities between Algorithm 1 and SISA [1]?**
>
> We provide a detailed comparison between SISA from Bourtoule et al. [3] and Algorithm 1 in Section A of our response to Reviewer sthQ.
>
> [3] https://arxiv.org/abs/1912.03817

---

### Official Review · Reviewer_r4sp · 2024-11-04

**Soundness:** 3
**Presentation:** 2
**Contribution:** 3
**Rating:** 6
**Confidence:** 3

**Summary:**

The paper proposes a new, system-aware formulation of machine unlearning, which takes into account the information that an attacker could recover by compromising the system. The authors develop exact system aware unlearning algorithms based on sample compression learning algorithms and establish the computation time, memory requirements, deletion capacity, and excess risk guarantees.

**Strengths:**

1. The idea of incorporating the observer's perspective into the unlearning process is novel.
2. The theoretical analysis is overall coherent.
3. The paper is well-structured.

**Weaknesses:**

1. Lack of quantitative comparisons with other methods (theoretically or empirically).
2. The writing of the intuition part in Section 2 is not clear enough. The reviewer is quite confused about the statements on lines 125-128.
3. Lack of definitions for notations, e.g., $\mathcal X, \mathcal Y, \mathcal Z, \mathcal Z^*, f^*(\cdot), \Delta(\mathcal F), \mathcal J$, etc.
4. The author is confused about the “core set deletion capacity $K$” and the "$K$ shards" in Algorithm 1. Are the two $K$s the same parameter?
5. Some writing issues: grammar mistakes and typos, e.g., "we developed algorithms for exact system aware unlearning algorithms" on line 526 and "exact system aware" on line 531.
6. The definition stated in Sekhari et al. (2021b) and Guo et al. (2019) is generally referred to as "*certified* machine unlearning".
7. Although this paper mainly focuses on the theoretical part, it would be better to include the experimental results in the main text.

**Questions:**

1. What is $e$ in $\frac{K}{e}$ on line 275?
2. What are $\mathcal T_P$ and $\mathcal T_f$ on line 6 of Algorithm 1?

---

> ### Author Response · Authors · 2024-11-22
> **Response to Reviewer r4sp**
>
> We thank the reviewer for their thoughtful feedback. We have addressed writing concerns and typos. We have uploaded an updated paper pdf which incorporates their feedback, and we address their comments below.
>
> > **Lack of quantitative comparisons with other methods (theoretically or empirically).**
>
> Exact unlearning algorithms under the previous definition of unlearning all involve some level of retraining from scratch for each deletion and storing the entirety of the dataset, which is expensive and inefficient in memory and time. Under our new definition, we can avoid this retraining; we can design algorithms that provably beat the time to retrain and provably require significantly less memory than the entirety of the dataset as long of effective core set algorithms exist. We indicate assumptions or natural conditions (like margin assumption) under which such core-set algorithms exist and so the comparison and improvement are straightforward. In the event no sample compression algorithm exists, our algorithm has the same complexity as existing exact unlearning algorithms. For an explicit comparison, we provide a detailed comparison between SISA from Bourtoule et al. [1] and Algorithm 1 in Section A of our response to Reviewer sthQ. Algorithm 2 is the first exact unlearning algorithm for linear classification that does not require the storage of the entire dataset so the comparison with previous algorithms is straightforward.
>
> > **The writing of the intuition part in Section 2 is not clear enough. The reviewer is quite confused about the statements on lines 125-128.**
>
> The motivating example in lines 125-128 illustrates that there exist unlearning algorithms that satisfy system-aware unlearning but do not satisfy traditional definitions of unlearning. Consider an algorithm that selects a few important points to train a model on: if one of the original important points were deleted, then perhaps the algorithm would have selected a different point that it deemed “important” to include in the model instead. Previous definitions of unlearning require including that new important point which in turn requires access to the entire dataset. In our case, we can simply ignore that point under system aware unlearning. The example in lines 125-128 is the simplest example of this. A more concrete example is the selective sampling case discussed in depth in Section 4.
>
> > **The author is confused about the “core set deletion capacity $K$” and the "$K$ shards" in Algorithm 1. Are the two $K$s the same parameter?**
>
> These two $K$s are the same parameter. The algorithm designer specifies the core set deletion capacity $K$ that they desire, and then Algorithm 1 divides the sample into $K$ shards accordingly, in order to ensure a core set deletion capacity of $K$.
>
> > **Although this paper mainly focuses on the theoretical part, it would be better to include the experimental results in the main text.**
>
> Our primary contribution is the new definition of unlearning which accounts for the attacker’s knowledge. We prove that under the assumption of a weaker adversary, we can design exact unlearning algorithms that are significantly more memory and computationally efficient. In the final version of the paper, we will bring the experiments to the main body.
>
> > **What is $e$ in $K/e$ on line 275?**
>
> The $e$ here refers to the mathematical constant, approximately 2.7182…
>
> > **What are $\mathcal{T}_P$ and $\mathcal{T}_f$ on line 6 of Algorithm 2?**
>
> We have clarified the definition of these two terms in line 5 of Algorithm 2 in the updated paper pdf.
>
> [1] https://arxiv.org/abs/1912.03817

---

> > ### Comment · Reviewer_r4sp · 2024-11-26
> >
> > Thank you for your response. Please address all the concerns (not just Q2) in the updated paper and upload the revised version.

---

> > > ### Author Response · Authors · 2024-11-26
> > > **Follow Up Response to Reviewer r4sp**
> > >
> > > We have uploaded a new PDF incorporating all of the reviewer's feedback. Below are pointers to the changes in the PDF.
> > >
> > > > **W1. Lack of quantitative comparisons with other methods (theoretically or empirically).**
> > >
> > > We have added a comparison with previous works in **lines 83-90**, highlighting our improvements and contributions.
> > >
> > > > **W2. The writing of the intuition part in Section 2 is not clear enough...**
> > >
> > > We have rewritten and clarified the description of our motivating example in **lines 121-133**.
> > >
> > > > **W3. Lack of definitions for notations...**
> > >
> > > We have clarified our notation in **lines 98-101** and adjusted our notation in **lines 143-144**.
> > >
> > > > **W4. The author is confused about the “core set deletion capacity $K$" and the "$K$ shards" in Algorithm 1...**
> > >
> > > We have added an explanation clarifying the connection between these two values in **lines 265-266**.
> > >
> > > > **W5. Some writing issues: grammar mistakes and typos...**
> > >
> > > We have corrected the typos in lines 526 and 531 (see the changes in **lines 532 and 536**) along with other writing issues.
> > >
> > > > **W6. The definition stated in Sekhari et al. (2021b) and Guo et al. (2019) is generally referred to as "certified machine unlearning".**
> > >
> > > We have incorporated this change in **line 107**.
> > >
> > > > **W7. Although this paper mainly focuses on the theoretical part, it would be better to include the experimental results in the main text."**
> > >
> > > In the final version of the paper, we will bring the experiments to the main body.
> > >
> > > > **Q1. What is $e$ in $\frac{K}{e}$ in line 275?**
> > >
> > > We clarify this **line 283**.
> > > > **Q2. What are $\mathcal{T}_P$ and $\mathcal{T}_f$$ on line 6 of Algorithm 2?**
> > >
> > > We have clarified the definition of these two terms in **line 5 of Algorithm 2**.
> > >
> > > We thank the reviewer again for their valuable feedback in helping us improve our paper. We are happy to address any additional concerns.

---

### Official Review · Reviewer_sthQ · 2024-11-10

**Soundness:** 3
**Presentation:** 3
**Contribution:** 2
**Rating:** 5
**Confidence:** 3

**Summary:**

The paper proposes system-aware unlearning which ensures that an observer cannot distinguish between a model trained on the initial dataset with unlearning applied and a model trained on a smaller dataset without the deleted points.

**Strengths:**

Sharding and sub-sampling can improve efficiency of unlearning algorithms

**Weaknesses:**

The paper ignores Bourtoule et al.

The paper needs to properly analyze the threats against prior unlearning algorithms (inference attacks against sequence of released models, revealing what data is deleted between two releases).

**Questions:**

A. The paper does not talk about the prior sharding methods (Bourtoule et al, Machine Unlearning). How does this work compare with that method (which is splitting data into shards, training one model on each shard, and retraining only the model that includes the to-be-deleted datapoint). I suggest the authors to comment on the fundamental differences between these works, and also comment on the followings.

1. Differences in computational efficiency during unlearning
2. Memory requirements for storing models/data
3. Impact on model accuracy as deletion requests increase
4. Scalability to large datasets or high-dimensional data


B. Many follow-up papers of Bourtoule et al highlighted the possibility of an adversary observing multiple snapshots of the models (over time, as unlearning requests are processed). Can you analyze the proposed system-aware unlearning algorithm in presence of adversaries with continuous observations of models? In particular, I suggest the authors to respond to the following questions.

1. How do the privacy guarantees of the proposed system-aware unlearning change under an adversary with continuous model observations?
2. Can you provide a comparative analysis of information leakage between your approach and previous methods in this scenario?
3. Are there potential modifications to your algorithm that could strengthen its resilience against such adversaries?
4. How does the computational overhead of maintaining privacy change in this threat model?

C. What are the limitations of the proposed method? Which threats remain unaddressed, and which types of algorithms are incompatible with this approach?

---

> ### Author Response · Authors · 2024-11-22
> **Response to Reviewer sthQ**
>
> We thank the reviewer for their thoughtful feedback. We have uploaded an updated paper pdf which incorporates their feedback, and we address their comments below.
>
> > **A. Fundamental differences between this work (Algorithm 1) and SISA from Bourtoule et al. - differences in computational efficiency, memory requirements, model accuracy, etc.**
>
> SISA from Bourtoule et al. [4] uses sharding and the storage of intermediate models as a means of speeding up retraining from scratch. Algorithm 1 requires no retraining or computation at the time of deletion. Let $n$ be the size of the sample and let $K$ be the number of shards. In the worst case, SISA requires retraining on a sample of size $n/K$ after each deletion. Algorithm 1 processes each deletion in time $O(1)$; no recomputation or retraining needs to be done at the time of unlearning. Furthermore, SISA requires the storage of all $n$ samples. Let $\mathfrak{C}(n/K)$ be the number of samples in the core set of each shard. Algorithm 1 requires the storage of $K \cdot \mathfrak{C}(n/K) << n$ samples, where $\mathfrak{C}(n/K)$ can be exponentially smaller than $\mathfrak{C}(n/K)$ [1]. SISA requires the storage of $R \cdot K$ models where $R$ is the number of slices within each shard, while Algorithm 1 only requires the storage of $K$ models. As the size of the dataset increases, the memory savings from sample compression in Algorithm 1 become more advantageous over SISA. With regards to model accuracy, SISA provides no theoretical guarantees of accuracy after deletion. In particular, deletion sequences that depend on the label of a data point will quickly degrade the accuracy of SISA; however, Algorithm 1 is robust against such deletion sequences. In general, until the number of core set deletions exceeds $K$, Algorithm 1 is able to maintain model accuracy guarantees.
>
> > **B. Can you analyze the proposed system-aware unlearning algorithms in the presence of adversaries with continuous observations of models?**
>
> Algorithm 1 is robust to adversaries with continuous observations of models because the models before and after deletion are trained on two independent sets of data, so an adversary that has observed the model before and after deletion would be unable to compute the difference and recover the deleted individual. A core set deletion under Algorithm 2 does leak information under an adversary with continuous model observations. Privacy can be preserved by adding appropriately calibrated noise to the model output after unlearning a core set point [2], which is computationally inexpensive. Note that noise only needs to be added for core set deletions; for traditional unlearning algorithms, noise needs to be added after every deletion.
>
> > **C. What are the limitations of the proposed method? Which threats remain unaddressed, and which types of algorithms are incompatible with this approach?**
>
> The primary unaddressed threat is when the attacker has access to the trace of the system over time over the course of learning (rather than simply the state of the system after unlearning). In this case, system-aware unlearning fails; however, it is unclear how an attacker would gain access to this trace if it is not stored by the system (or compromised otherwise). Algorithms that can learn well with a small number of points are most compatible with this approach, such as selective sampling and sample compression algorithms which have theoretical upper bounds on the number of points required to learn a good classifier [1]. However, we can expand this framework beyond sample compression algorithms: after training on the full sample, one can use influence functions or data attribution [3] to identify points that have a small influence on the model, remove these points, and train a model on the smaller set of impactful points. This gives a more general approach to system-aware unlearning. Furthermore, unlearning algorithms under previously proposed definitions continue to be valid under system-aware unlearning, and system-aware unlearning allows for a new class of algorithms (ie. sample compression algorithms) to be easily compatible with unlearning.
>
> [1] https://arxiv.org/abs/2307.04998
>
> [2] https://arxiv.org/pdf/2106.04378
>
> [3] https://arxiv.org/pdf/2410.23232
>
> [4] https://arxiv.org/abs/1912.03817

---

> > ### Author Response · Authors · 2024-12-03
> > **Response to Reviewer sthQ**
> >
> > We provide an empirical comparison between SISA from Bourtoule et al [1] and our Algorithm 1 on two binary classification datasets: the Purchase dataset (249,215 points, dimension = 600) provided by Bourtoule et al [1] and our dataset (200,000 points, dimension = 100, margin = 0.1) used in our experiments in Appendix A.1. On these two datasets, we compared the performance of Algorithm 1 using selective sampling for sample compression and the performance of SISA under a sequence of 80,000 label-dependent deletions. We plot the resulting model accuracy over the course of these deletions for both algorithms, and we compare the memory and computation time. We summarize the results below.
> >
> > **Purchase dataset from Bourtoule et al [1]**
> >
> > Plot of model accuracy of Algorithm 1 vs SISA over the course of 80,000 label-dependent deletions: https://anonymousauthors83274.github.io/plots/
> >
> > | | Initial training time (secs) | Total accumulated deletion time (secs) | Percent of data stored in memory (%) |
> > | -------- | ------- | -------- | ------- |
> > | Algorithm 1 [ours] | 206.6 | <1 | 39.2% |
> > | SISA [1] | 30.2 | 1174.3 | 100% |
> >
> > **Main Takeaways:** The initial accuracy of Algorithm 1 is slightly worse than SISA, but Algorithm 1 is able to maintain significantly better accuracy under a longer sequence of label dependent deletions compared to SISA. Algorithm 1 requires longer initial training time, but Algorithm 1 requires significantly less computation time at the time of deletion. Furthermore, Algorithm 1 requires the storage of significantly less samples.
> >
> > **Our dataset from Appendix A.1**
> >
> > Plot of model accuracy of Algorithm 1 vs SISA over the course of 80,000 label-dependent deletions: https://anonymousauthors83274.github.io/plots/
> >
> > | | Initial training time (secs) | Total accumulated deletion time (secs) | Percent of data stored in memory (%) |
> > | -------- | ------- | -------- | ------- |
> > | Algorithm 1 [ours] | 1.7 | <1 | 1.3% |
> > | SISA [1] | 20.6 | 697.1 | 100% |
> >
> > **Main Takeaways:** When the dataset allows for a more favorable compression scheme, as this dataset does, Algorithm 1 is able to match the initial accuracy of SISA, despite using much less data, and Algorithm 1 is able to maintain significantly better accuracy under a longer sequence of label dependent deletions. Furthermore, Algorithm 1 requires significantly less memory and significantly less computation time, both at the time of training and the time of deletion, due to increased sample compression. When the dataset allows for significant compression, Algorithm 1 dominates SISA in accuracy, memory, and computation time.
> >
> > These empirical results support our theoretical analysis. We will add both the theoretical and empirical comparison of Algorithm 1 and SISA to the final version of the paper. We appreciate the reviewer's valuable feedback.
> >
> > [1] https://arxiv.org/abs/1912.03817

---

### Meta-Review · Area_Chair_5QZE · 2024-12-20

**Metareview:**

The reviewers were mixed about the paper. While there were support for the paper, still a couple of expert reviewers in the field did not show support for the paper. The main concerns were:

- Lack of quantitative comparisons with other methods (theoretically or empirically).
- Necessity of "system aware unlearning" is not well justified.
- Lack of empirical evidence to support the claim that Algorithm 1 is able to maintain much better accuracy compared to algorithms like -
 - SISA from [1] under deletion sequences which are label dependent and cause a large distribution shift.

**Additional Comments On Reviewer Discussion:**

See above.

---

### Decision · Program_Chairs · 2025-01-22

Reject